



# Specified dynamics scheme impacts on wave-mean flow dynamics, convection, and tracer transport in CESM2 (WACCM6)

Nicholas A. Davis [1], Patrick Callaghan [2], Isla R. Simpson [2], and Simone Tilmes [1]

[1]Atmospheric Chemistry Observations and Modeling Laboratory, National Center for Atmospheric Research, Boulder, CO, USA

[2]Climate and Global Dynamics Laboratory, National Center for Atmospheric Research, Boulder, CO, USA

**Correspondence:** Nicholas A. Davis (nadavis@ucar.edu)

**Abstract.** Specified dynamics schemes are ubiquitous modeling tools for isolating the role of dynamics and transport on chemical weather and climate. They typically constrain the circulation of a chemistry-climate model to the circulation in a reanalysis product through linear relaxation. However, recent studies suggest that these schemes create a divergence in chemical climate and the meridional circulation between models and do not accurately reproduce trends in the circulation. In this study

we perform a systematic assessment of the specified dynamics scheme in the Community Earth System Model version 2, Whole Atmosphere Community Climate Model version 6 (CESM2 (WACCM6)), which proactively nudges the circulation toward the reference meteorology. Specified dynamics experiments are performed over a wide range of nudging timescales and reference meteorology frequencies, with the model's circulation nudged to its own free-running output - a clean test of the specified dynamics scheme. Errors in the circulation scale robustly and inversely with meteorology frequency, and have little

dependence on nudging timescale. However, the circulation strength and errors in tracers, tracer transport, and convective mass flux scale robustly and inversely with nudging timescale. A 12 to 24 hour nudging timescale at the highest possible reference meteorology frequency minimizes errors in tracers, clouds, and the circulation, even up to the practical limit of one reference meteorology update every timestep. The residual circulation and eddy mixing integrate tracer errors and accumulate them at the end of their characteristic transport pathways, leading to elevated error in the upper troposphere/lower stratosphere and in

the polar stratosphere. Even in the most ideal case, there are non-negligible errors in tracers introduced by the nudging scheme. Future development of more sophisticated nudging schemes may be necessary for further progress.

## 1 Introduction

Anthropogenic and natural emissions of gases and aerosols have substantial human and ecological consequences by virtue of

atmospheric transport. In the troposphere, sporadic convection in the tropics rapidly lofts boundary layer air to high altitudes where it can slowly ascend through the tropical tropopause layer into the middle atmospheric Brewer-Dobson circulation (Fueglistaler et al., 2009; Butchart, 2014). Some of this air is swept through the shallow branch of the circulation, where it





quickly returns to the troposphere (Birner and Bönisch, 2011; Abalos et al., 2013; Garny et al., 2014). But for the air that ascends up through the depth of the stratosphere, its fate is tied to the tug-of-war of the seasons (Ploeger and Birner, 2016).

Throughout the annual cycle the residual circulation reverses course from north to south, to south to north, sloshing the air back and forth on a long, multi-year journey to the poles where it finally seeps back down to the troposphere. Mixing by breaking waves recirculates some of this air, increasing its stratospheric residence time beyond what would be predicted by residual circulation trajectories alone (Garny et al., 2014). In the troposphere, air is rapidly mixed throughout the extratropics (Waugh et al., 2013; Yang et al., 2019), with nearly half of the residual circulation mass transport occurring via moist diabatic ascent in

the storm tracks (Pauluis et al., 2008).

This global transport is especially important in the context of halogens, radiatively-active aerosols like black carbon and sulfates, and health-relevant species including ground-level particle matter, ozone, and ozone precursors including carbon monoxide. Oceanic emissions of reactive chlorine and bromine species mediate tropospheric ozone (Yang et al., 2005) and can be transported up to the stratosphere where they contribute to spring ozone loss (Daniel et al., 1999; Salawitch et al., 2005;

Sinnhuber et al., 2009; Hossaini et al., 2017). Additionally, anthropogenic emissions of CFC-11 in apparent violation of the Montreal Protocol have been transported up to the stratosphere and have delayed the recovery of the ozone hole (Montzka et al., 2018; Dhomse et al., 2019). In the troposphere, long-range transport of black carbon aerosols can have profound impacts on Arctic climate (Shindell et al., 2008), and may be a major driver of observed tropical expansion (Allen et al., 2012; Kovilakam and Mahajan, 2015; Zhao et al., 2020). However, transport can occur along different pathways depending upon whether the

emissions occur over land or ocean, and can be especially sensitive to the configuration of the large-scale atmospheric circulation (Yang et al., 2019). Recent work suggests that transport processes make substantial contributions to hazardous peaks in summertime ozone (Kerr et al., 2019), and that variations in the latitude of the tropospheric jet project directly onto surface ozone (Barnes and Fiore, 2013; Kerr et al., 2020).

The particular nature of this transport is difficult to quantify because its characteristics span multiple orders of magnitude in

time and space. A recent exchange in the literature illustrates the degree to which circulation uncertainty can lead to opposing conclusions. Ball et al. (2018) reported that two chemistry-climate models forced by reanalysis meteorology in so-called "specified dynamics" configurations were unable to reproduce the recently observed decline in lower stratospheric ozone. If true, it would open up roles for novel chemistry or emissions of ozone depleting substances. Shortly thereafter, Chipperfield et al. (2018) reported that a chemical transport model was able to reproduce the ozone trends when using the same meteorology.

Additionally, Wargan et al. (2018) reported that a replay model simulation could reproduce most of the lower stratospheric ozone loss using reanalysis meteorology, though not in the deep tropics.

Specified dynamics schemes are one flavor of modeling techniques that constrain known circulation variability to isolate the role of the atmospheric circulation in driving chemical weather and climate. They are also important tools for evaluating different chemical and physics schemes, interpreting field campaign observations, and performing chemical forecasts. Specified

dynamics schemes typically consist of a linear relaxation of fields such as temperature and horizontal winds to a reference meteorology, which is almost always a reanalysis product. More sophisticated techniques include NASA's replay, in which the model is replayed after an initial integration with the tendencies needed to match the reference meteorology (Orbe et al.,





2017; Wargan et al., 2018). A chemical transport model is wholly different. While it will include parameterizations of physical processes, it is devoid of a true dynamical core as the reference meteorology is simply ingested directly (see, for a relevant

example, Chipperfield (1999, 2006)). The exchange in the aftermath of Ball et al. (2018) indicates that one of more of these methods is deficient.

Kinnison et al. (2007) showed that tracer distributions in a chemical transport model were highly sensitive to arguably minor differences in the circulation across three reference meteorologies. Analyses of the models in the Chemistry-Climate Model Intercomparison (CCMI) indicate that the intermodel spread in the meridional circulation in specified dynamics simulations

is as large or larger than the intermodel spread in free-running simulations (Orbe et al., 2018; Chrysanthou et al., 2019). Unfortunately, the different models in CCMI use different specified dynamics techniques and reference meteorologies, so it is difficult to isolate the cause of this divergence in the circulation.

There is evidence that the relaxation timescale impacts transport (Merryfield et al., 2013; Orbe et al., 2017; Hardiman et al., 2017). Some studies explicitly nudge the rotational part of the flow on a faster timescale than the divergent part of the

flow (Löffler et al., 2016; van Aalst et al., 2004), the goals being to constrain the more certain aspect of the circulation and to allow the model physics the freedom to dictate the divergent part of the flow. Temperature nudging acts like diabatic heating and modulates the strength meridional circulation, leading to systematic errors in the circulation and in tracers such as ozone (Miyazaki et al., 2005; Akiyoshi et al., 2016). More exotic nudging techniques, such as climatological anomaly (Zhang et al., 2014) and zonal anomaly nudging (Davis et al., 2020), have revealed that the nudging of zonal mean temperatures can be a

major source of error. However, it is often desirable to nudge temperatures to ensure that temperature-dependent chemistry in the model is consistent with the real atmosphere (Solomon et al., 2015, 2016; Froidevaux et al., 2019). We cannot just forsake temperature nudging.

In this study, we perform a clean test of a the specified dynamics scheme in CESM2 (WACCM6) in which we nudge the model to reference meteorology created by itself. In this configuration, we fully eliminate errors and uncertainty associated

with using reference meteorology from a different modeling system, but also expand the possible phase space of our analysis to its practical limits. An exhaustive set of simulations in which both the nudging timescale and meteorology frequency are varied reveals coherent, global patterns of circulation error that project onto errors in stratospheric and tropospheric carbon monoxide and ozone. While we highlight one particular configuration that minimizes errors in the circulation, clouds, and constituents, substantial room for improvement remains and will likely require innovating beyond linear relaxation of the full meteorology.

## 85   2   Model configuration

The CESM2.0 (WACCM6) finite volume dynamical core (Gettelman et al., 2019) is run at 1 degree horizontal resolution with 110 vertical levels from the surface to approximately 140 km in the lower thermosphere for one year from January 1, 2018 to December 31, 2018. We run so-called "F" compset cases, with prescribed sea surface temperatures and sea ice and prescribed vegetation phenology based on observations. Solar forcings, greenhouse gas lower boundary conditions, volcanic emissions,

and other surface emissions are prescribed according to the CMIP6 SSP5-8.5 scenario (O'Neill et al., 2016). Anthropogenic



emissions of volatile organic, non-methane volatile organic, and other compounds are prescribed by the Copernicus Atmosphere Modeling Service 81 dataset (Granier et al., 2019), while fire emissions are prescribed by the Fire INventory from NCAR version 1 dataset (Wiedinmyer et al., 2011). The quasi-biennial oscillation is not prescribed, but is instead spontaneously driven by internally-generated waves (Garcia and Richter, 2019). All simulations have identical initial conditions
derived from the same spin-up run on January 1, 2018.

There are two specified dynamics implementations in CESM2. "SD" compsets are configured to apply nudging only within the finite volume dynamical core as described in Kunz et al. (2011), with the applied tendencies controlled via the "met" namelist (see section 6.4 of the CAM6 user's guide, url in the *Code and data availability* section). Here, however, we use the alternative dynamical-core-independent method that applies nudging as physics tendencies controlled via the "nudging"
namelist (see section 9.6 of the CAM6 user's guide). For a nudged variable, $x$, the nudging tendency is applied as a linear relaxation of the form

$$\frac{\partial x}{\partial t}_{sd} = -W\frac{(x - x_{ref})}{\tau}, \tag{1}$$

where $x_{ref}$ is the reference meteorology value at the next reference meteorology update step, $\tau$ is the relaxation timescale, and $W$ is a window function that limits the spatial domain over which the tendency is applied. In this formulation, the nudging
proactively pulls the model state toward the next instantaneous reference meteorology value. $W$ is set to 1 below 1 hPa and linearly tapers to 0 between 1 and 0.1 hPa to avoid numerical instabilities related to nudging in the presence of atmospheric tides and large gravity wave tendencies. In the simulations in this study we nudge the zonal wind, meridional wind, and temperature, although the model can be configured to nudge any combination of these variables.

In computing the tendencies there are three relevant stepping intervals: $N_{step} = 48$, the number of dynamical timesteps
per day, $N_{obs}$, the number of times per day that reference meteorology is available, and $N_{update}$, the number of times per day the nudging tendency in Eq. 1 is updated. We set $N_{update}$ to 48 so that the nudging tendency is updated every dynamics timestep with the most recent value of $x$ from the nudging simulation. Reanalysis output is typically provided every 6 hours ($N_{obs} = 4$) or 3 hours ($N_{obs} = 8$). Here, a free running simulation is used to generate reference meteorology every 30 minute dynamics timestep. A suite of specified dynamics simulations are then used to explore how variations in the nudging parameters
$N_{obs}$, the reference meteorology frequency, and $\tau$, the nudging timescale, shape the errors in the resulting simulation. This is advantageous because our analysis will not be obscured by differences in topography, physics, and dynamical cores if we were to nudge our model to reanalysis output. It will also allows us to explore the parameter $N_{obs}$ over a larger range of values up to the limit of a new reference meteorology value every timestep.

To validate the nudging method, a null test with a 30 minute nudging timescale in which the reference data is sampled at
every 30 minute model timestep ($N_{obs} = 48$) and the tendencies are updated at every timestep ($N_{update} = 48$) must return nudging tendencies that remain zero and model states that perfectly match the evolution of the reference meteorology. The method fails this null test in the current implementation of CESM2 because while the reference meteorology is generated at the end of the dynamics step and before the physics step, the nudging tendencies are applied at the end of the physics step when all other parameterizations have already modified the model state. If the physics order is modified so that the nudging





tendencies are calculated and applied at the start of the physics step, the model passes this null test. Future versions of CESM2 will contain this correction to the physics order.

We test six different nudging timescales: 2 hours, a very short timescale to constrain high frequency variability; 4 and 6 hours, short timescales to constrain sub-diurnal variability; 12 and 24 hours, moderate timescales to constrain diurnal variability; and 48 hours, a longer timescale to constrain synoptic variability. We also test five different meteorology frequencies: 1 update per

day, the arbitrary limit of the nudging scheme; 4 updates per day, equivalent to 6-hourly reanalysis meteorology; 8 updates per day, equivalent to 3-hourly reanalysis meteorology; 24 updates per day; and 48 updates per day, the practical limit of an update every timestep. Combinations of nudging parameters that require an increase in the "nsplit" parameter, the finite volume advection subcycling, to run without advection errors are considered here to be unstable. 2 and 4 hour nudging at 1, 4, and 8 updates per day, and 6, 12, and 24 hour nudging at 1 and 4 updates per day require an increase in advection subcycling, so they

are not considered viable configurations. They are generally cases in which the nudging timescale is close to or faster than the meteorology frequency. All other combinations are stable, resulting in 18 distinct specified dynamics simulations.

For temperature, convective mass flux, and ozone and carbon monoxide mole fraction we archive the output at every model timestep to ensure that our error analysis captures the full variability in each field. For wave-mean flow dynamics and transport we archive daily average fields, including daily average eddy fluxes of heat, momentum, and tracers. All eddy fluxes are cal-

culated online in the model every timestep, such that their daily average is the true average. See the *Code and data availability* section for information on the modifications to the source code necessary for this output.

While the El Niño-Southern Oscillation was generally neutral in 2018, the reference simulation generated a sudden warming on February 21, which was remarkably close to the observed sudden warming on February 18th and may impact the results for the Northern Hemisphere.

## 3   Methods

Model performance is assessed using three measures of disagreement: the root-mean-square temporal error (RMSTE), the root-mean-square spatial error (RMSSE), and the sign-adaptive mean error (SAME). The root-mean-square temporal error of a variable $x$ is defined as

$$RMSTE(p, \phi, \lambda) = \sqrt{\overline{(x_{s.d.}(t, p, \phi, \lambda) - x_{ref}(t, p, \phi, \lambda))^2}}, \tag{2}$$

where $t$, $p$, $\phi$, and $\lambda$ are the time, pressure, latitude, and longitude coordinates, the overbar indicates the time mean, $x_{s.d.}$ is the value of $x$ in the specified dynamics simulation, and $x_{ref}$ is the value of $x$ in the reference simulation. In some cases we vertically and meridionally average this error. The vertical and meridional average of some variable $x$ between two pressure levels $p_1$ and $p_2$, with $p_1 > p_2$, is given as

$$\langle x \rangle = \frac{1}{(p_1 - p_2)} \int\limits_{p=p_2}^{p_1} \left( \frac{1}{2} \int\limits_{\phi=-90}^{90} [x] \cos(\phi) d\phi \right) dp, \tag{3}$$





where $\phi$ is latitude and brackets indicate the zonal mean. Errors are averaged from 200 hPa to 1000 hPa for a tropospheric average, averaged from 1 hPa to 200 hPa for a stratospheric average, and averaged from 1 hPa to 1000 hPa for a global average. The root-mean-square spatial error is defined as

$$RMSSE(t) = \sqrt{\left\langle \left( [x_{s.d.}(t,p,\phi)] - [x_{ref}(t,p,\phi)] \right)^2 \right\rangle}. \tag{4}$$

The root-mean-square temporal error, which we will refer to hereafter as "temporal error", quantifies the average error in
temporal variability, while the root-mean-square spatial error, which we will refer to hereafter as "spatial error", quantifies the error in the zonal mean structure at each time step.

To quantify the mean bias for fields that are not single-signed, we modify the formula for the mean bias to account for the average sign of the field in the reference meteorology to derive the sign-adaptive mean error (SAME),

$$SAME(p,\phi,\lambda) = \left( \overline{x}_{s.d.}(p,\phi,\lambda) - \overline{x}_{ref}(p,\phi,\lambda) \right) \frac{\overline{x}_{ref}(p,\phi,\lambda)}{|\overline{x}_{ref}(p,\phi,\lambda)|}. \tag{5}$$

Negative values of the SAME indicate that the field is weaker in magnitude than it is in the reference simulation - up to and including opposite in sign. We will periodically display the actual mean bias and refer to both as "mean errors", but will always use the SAME to calculate averaged mean errors.

In principle, the spatial and temporal error can be corrected for the mean bias (see, for example, Murphy and Epstein (1989)). However, this would require a reasonable estimate of the long term climatology of this particular model configuration,
for which we only have a single year. While our discussion will address the spatial, temporal, and mean error as separate terms, it should be noted that they are not orthogonal.

## 3.1 Transformed Eulerian Mean dynamics

We will use the transformed Eulerian mean (TEM) framework to more directly delineate the relationships between errors in eddy-mean flow dynamics and chemical transport. The TEM zonal mean zonal momentum equation in log-pressure coordinates
is given by

$$\frac{\partial [u]}{\partial t} = \frac{1}{\rho_0 a \cos(\phi)} \nabla \cdot \boldsymbol{F} - \frac{g}{2\pi \rho_0 a \cos^2(\phi)} \nabla \cdot ((\boldsymbol{\nabla} \times \boldsymbol{\Psi}^*)[M]) + [X], \tag{6}$$

where $u$ is the zonal wind, $F$ is the Eliassen-Palm (EP) flux vector, $\Psi^*$ is the TEM residual streamfunction, $M$ is the angular momentum per unit mass, $X$ is a catch-all for non-conservative forces including friction, $a$ is the radius of the earth, and $\nabla$ is the zonal mean divergence operator in spherical coordinates given as

$$\nabla = \left\{ \frac{1}{a\cos(\phi)} \frac{\partial}{\partial \phi} \cos(\phi), \frac{\partial}{\partial z} \right\}. \tag{7}$$

The log-pressure height $z$ is given by the transformation $z = -H \ln(p/p_r)$, where $p$ is the pressure, $p_r = 1000$ hPa is the reference surface pressure, and the scale height $H$ is taken as 6800 m. Log-pressure density $\rho_0$ is given by $\rho_0 = \rho_r(p/p_r) = p/(Hg)$, where $g$ is the acceleration due to gravity.





The EP flux is

$$\boldsymbol{F} = \left\{ F^{\phi}, F^{z} \right\} = \left\{ \rho_0 \left( \frac{\partial [M]}{\partial z} \frac{[v'\theta']}{\partial \theta / \partial z} - [M'v'] \right), \rho_0 \left( \frac{1}{a} \frac{\partial [M]}{\partial \phi} \frac{[v'\theta']}{\partial [\theta] / \partial z} - [M'w'] \right) \right\}, \tag{8}$$

where $v$ and $w$ are the meridional and vertical velocities, $\theta$ is the potential temperature, and primes denote deviations from the zonal mean. The EP flux corresponds to the group velocity of steady, linear Rossby waves, and traces Rossby wave propagation in the meridional plane (Edmon et al., 1980). By virtue of their intrinsic easterly phase speeds, Rossby waves gather easterly momentum from their source regions and deposit easterly momentum where they dissipate, respectively leading to acceleration of and drag on westerly flow.

The TEM residual streamfunction is given by

$$\Psi^* = \frac{2\pi a \cos(\phi)}{g} \int\limits_0^p [v^*] dp, \tag{9}$$

where $[v^*]$ is the TEM residual circulation meridional velocity,

$$[v^*] = [v] - \frac{\partial}{\partial z} \left( \frac{[v'\theta']}{\partial \theta / \partial z} \right). \tag{10}$$

As it is equivalent to the Eulerian mean flow in the absence of meridional eddy heat fluxes, the TEM residual circulation can be interpreted as that part of the meridional flow with the meridional-eddy-heat-flux-induced adiabatic recirculations removed. This can also be deduced from the equivalence between the TEM meridional circulation and the Eulerian mean meridional circulation in isentropic coordinates (Juckes, 2001). It is therefore a "quasi Lagrangian" approximation of zonal mean parcel trajectories, which gives it greater utility for examining transport than the Eulerian mean.

## 3.2 Chemical transport

Chemical transport is assessed with the TEM transport equation,

$$\frac{\partial [\chi]}{\partial t} = \frac{1}{\rho_0} \nabla \cdot \boldsymbol{F}_{\boldsymbol{\chi}} - (\nabla \times \Psi^*) \cdot \nabla [\chi] + [S], \tag{11}$$

where $\chi$ is the mole fraction of some chemical species, $S$ is its source/sink, and $\boldsymbol{F}_{\boldsymbol{\chi}}$ is the TEM eddy transport vector given by

$$\boldsymbol{F}_{\boldsymbol{\chi}} = \left\{ F_{\chi}^{\phi}, F_{\chi}^{z} \right\} = \left\{ \rho_0 \left( \frac{\partial [\chi]}{\partial z} \frac{[v'\theta']}{\partial [\theta] / \partial z} - [v'\chi'] \right), \rho_0 \left( \frac{1}{a} \frac{\partial [\chi]}{\partial \phi} \frac{[v'\theta']}{\partial [\theta] / \partial z} + [w'\chi'] \right) \right\}. \tag{12}$$

It is not coincidental that this has the same form as the EP flux vector, as that is simply the TEM eddy transport vector for angular momentum. The characteristic feature of the TEM is that for all tracers, the adiabatic transport by the mean flow induced by the meridional eddy heat fluxes is absorbed into the eddy transport itself.

## 4 Global errors in meteorology

Temporal (Fig. 1) and spatial (Fig. 2) errors in temperature, EP flux divergence, and the TEM streamfunction decrease exponentially with increasing meteorology frequency and decrease slightly with decreasing nudging timescale. Recall that nudging





nudging timescale refers to the linear relaxation timescale in Eq. 1, while meteorology frequency refers to the number of times per day that the reference meteorology is updated in the specified dynamics simulation. The errors asymptote toward the maximum possible meteorology frequency of 48 per day – exposing a lower limit of 10% temporal error and 0.1-10% spatial error relative to the temporal and spatial variability. Mean errors do not scale with meteorology frequency at all, and instead decrease

with increasing nudging timescale (Fig. 3; if one neglects the combination of 48 hour/1 per day, which is a major outlier for the circulation metrics). For temperature and convective mass flux, both thermodynamic quantities, the mean error disappears at longer timescales (Fig. 3a,b), though for the circulation the mean errors reach a minimum at a 24 hour timescale (Fig. 3c,d). Temperature, the EP flux divergence, and the TEM streamfunction become biased too high by increasing nudging timescale, while convective mass flux becomes biased too weak.

Temporal and spatial errors in the convective mass flux decrease exponentially with increasing meteorology frequency, but reach a minimum at a 12 hour nudging timescale (Fig. 1b, Fig. 2b). At shorter and longer nudging timescales the error increases. For parameterized processes such as clouds, nudging presents a conundrum. Cloud heating will be built in to the reference meteorology temperature, so that nudging to the reference meteorology temperature will effectively perform some fraction of the convective heating via the nudging itself. It is indeed the case that the global convective mass flux erroneously weakens

with decreasing nudging timescale (Fig. 3b), which in the tropics manifests as a robust decrease below 500 hPa (Fig. 4a,b). There are time-average positive temperature nudging tendencies at the characteristic altitudes of shallow and deep convective heating that may be acting to suppress convection (Fig. 4c). Aloft, negative temperature nudging tendencies that scale with decreasing nudging timescale are associated with increased convective mass flux, which may be acting in lieu of cloud-top radiative cooling that would otherwise occur. Nudging, especially at a timescale shorter than 24 hours, incurs substantial and

apparently unavoidable (Fig. 3b) penalties in convective mass flux.

On the other hand, nudging at too long or short a timescale leads to clouds occurring at different times (Fig. 1b) and in different places (Fig. 2b) than the reference meteorology, even though it may result in minimal global mean error (Fig. 3b). While spatial and mean errors in the convective mass flux asymptote at approximately 10% of the total variability, temporal errors asymptote at values equal to and larger than the variability.

The positive mean error of the EP flux divergence, indicating generally greater wave generation and wave drag, is generally consistent with the positive mean error in the wave-driven TEM streamfunction (Fig. 3c,d). The tropical stratospheric upwelling velocity is an especially important measure of the residual circulation, as it diagnoses the net transport through the tropical tropopause layer and into the stratosphere. Mean tropical upwelling in the lower stratosphere increases consistently with decreasing nudging timescale, with a peak increase of 3-5% at a 2 hour nudging timescale (Fig. 5).

## 5  Global errors in tracers

Errors in ozone and carbon monoxide mole fraction in the troposphere and stratosphere behave similarly to errors in the convective mass flux (Fig. 6, 7), with a minimum in spatial and temporal errors at 12 to 24 hour nudging timescales. While temporal errors are high at only the shortest and longest nudging timescales, spatial errors are especially high at nudging





timescales shorter than 12 hours. The temporal errors asymptote at around 10-25% of the total variability, while spatial errors

asymptote at a mere 0.1-5%. This is probably due to the first-order influence of photochemistry on the global distribution of these tracers. Spatial error is a relatively strong function of timescale, while temporal error tends to scale more consistently with meteorology frequency.

The mean errors in tropospheric/stratospheric carbon monoxide and stratospheric ozone mole fraction decrease with increasing nudging timescale (Fig. 8a,b,d), and all but disappear at a 48 hour nudging timescale. In both regions, carbon monoxide is

biased high, while in the troposphere ozone is biased low. Stratospheric ozone displays a unique dependence on meteorology frequency, although it is still also governed by nudging timescale and is biased high, unlike tropospheric ozone (Fig. 8c). As for the spatial and temporal errors, the mean error in stratospheric ozone also reaches a minimum at the 12-24 hour nudging timescale. However, the mean errors are generally small in all cases, ranging from 5% for stratospheric carbon monoxide to 0.1% for stratospheric ozone.

These mean errors in the troposphere are surprising given the known influence of convective mass transport. Deep convection rapidly transports boundary layer air to the upper troposphere. This air is relatively low in ozone and relatively rich in carbon monoxide, such that convection acts to reduce upper tropospheric ozone (Folkins et al., 2002; Doherty et al., 2005; Voulgarakis et al., 2009; Paulik and Birner, 2012) and increase upper tropospheric carbon monoxide (Kar et al., 2004; Park et al., 2009). The weakening convective mass flux with decreasing nudging timescale (Fig. 4) should lead to elevated free tropospheric

ozone and reduced free tropospheric carbon monoxide in the tropics. However, the opposite occurs, with reduced ozone and increased carbon dioxide throughout the whole troposphere and the lower stratosphere (Fig.'s 9, 10). An alternative explanation is that the acceleration of the residual circulation with decreasing nudging timescale leads to anomalously negative ozone and anomalously positive carbon monoxide advection tendencies in the free troposphere, which propagate up though the tropical pipe (Fig. 9c, 10c). It is unclear why the broader but slower ascent by the residual circulation would exert a more dominant

control than localized but more intense convective mass transport. It may be that changes in convective transport are more readily damped by horizontal advection than changes in zonal mean ascent.

The serious impact of nudging timescale and meteorology frequency on ozone and carbon monoxide warrants further investigation. Errors in the tropics seem consistent with circulation differences rather than convective transport differences, so it seems reasonable to posit that differences in the resolved circulation are the dominant source of the error.

## 6   Errors in eddy-mean flow dynamics

The systematic variation of temporal, spatial, and mean errors with nudging timescale and meteorology frequency means that they can be reliably described by appropriate regressions across the parameter phase space. While temporal and spatial errors in the circulation appear to be governed primarily by meteorology frequency (Fig. 1, 2), temporal and spatial errors in ozone and carbon monoxide appear to be governed by meteorology frequency and nudging timescale, respectively (Fig. 6, 7). Mean

errors in all fields are primarily governed by nudging timescale (Fig. 3, 8). Therefore, we will focus on the regression of TEM streamfunction and EP flux divergence spatial and temporal errors on meteorology frequency, while we will focus on





the regression of ozone and carbon monoxide temporal and spatial errors on meteorology frequency and nudging timescale, respectively. For all fields, we will focus on the regression of mean error on nudging timescale. We will display the negative of all error regressions for ease of discussion.

To examine the structure of temporal and mean errors, we simply project the zonal mean root-mean-square temporal error or mean error in each simulation onto either the logarithm of meteorology frequency or onto a one standard deviation change in nudging timescale (about 15 hours). For examining spatial errors, we first project each field onto the time series of its spatial error to obtain a zonal mean map, and then project the maps from each simulation onto either the logarithm of meteorology frequency or onto nudging timescale. One can interpret the first regression as the change in temporal or mean error per change in either meteorology frequency or nudging timescale. The second regression is more nuanced, as it is not the spatial error that

is regressed but instead the pattern in the physical field associated with variations in spatial error. The second regression should therefore be interpreted as the (erroneous) pattern in the physical field associated with either meteorology frequency or nudging timescale. It is useful only as a visualization of the structures that produce spatial error and vary with nudging parameters – and not indicative of the change in the spatial error itself, which only has the dimension of time.

As meteorology frequency decreases, the temporal error in the TEM streamfunction increases in general proportion to the climatology (Fig. 11a). In the troposphere, temporal errors in the midlatitude Ferrel circulation, but not the Hadley cell, increase with decreasing meteorology frequency (Fig. 11a). The projection of TEM streamfunction spatial error is characterized by a single pole-to-pole, surface-to-stratopause cell (Fig. 11c), which suggests that the errors arise when the solsticial residual circulation is generally too strong and expansive in one hemisphere, and too weak and contracted in the other. In the troposphere,

spatial error is especially concentrated in the tropics and the storm tracks where there is moist adiabatic ascent.

As meteorology frequency decreases the temporal errors in the stratospheric EP flux divergence increase poleward of the climatological divergence and deep within the polar vortices, associated with errors in the meridional propagation of Rossby waves (Fig. 11b). In the upper stratosphere the meridional position of the peak EP flux divergence dominates the spatial error in the Northern Hemisphere, while it is instead dominated by the total magnitude of the EP flux divergence in the Southern

Hemisphere (Fig. 11d). In both hemispheres, this error is governed by a vertical redistribution of wave activity, rather than by changes in meridional propagation. In the troposphere, temporal errors in the EP flux divergence increase with decreasing meteorology frequency in the extratropics in proportion to the climatology. There is no planetary-scale structure to the spatial error in the troposphere, although it appears roughly antisymmetric about the equator (Fig. 11d). As in the stratosphere, this error is governed primarily by a vertical redistribution of wave activity.

In the troposphere the TEM streamfunction and EP flux divergence/convergence are invigorated by decreasing nudging timescale (Fig. 11e,f). The invigoration of the stratospheric TEM streamfunction is limited to the shallow branch of the circulation, with error rapidly tapering off above 50 hPa. Global average mean errors in the circulation are therefore reflective of the mean error almost everywhere in the troposphere, but virtually nowhere in the stratosphere (Fig. 3c,d).

The physical coupling between the wave-driven TEM streamfunction and EP flux divergence raises the question of which

aspect of the circulation – the zonal mean or the eddies – is the source of their errors (Fig. 1c,d; 2c,d). A simple diagnostic is to "swap" either the zonal mean or eddy fields from the reference meteorology into the calculation of the TEM streamfunction





and EP flux divergence in the nudged simulations and recalculate the errors. The reduction in error between the swapped and non-swapped simulations measures the impact of the swapped field on the error. It is only diagnostic, and does not entertain any feedbacks between the eddies and the mean flow.

Temporal, spatial, and mean errors in the TEM streamfunction are overwhelmingly due to the Eulerian-mean part of the circulation, while the errors in the EP flux divergence are entirely due to the eddy fields (Fig. 12). The eddy fluxes that comprise the EP flux divergence are merely scaled by their projection onto angular momentum and static stability, so it isn't so surprising that the Eulerian mean contribution is negligible. It does seem surprising that the Eulerian mean dominates the errors in the TEM streamfunction, though. While the correction for the eddy recirculations is not a dominant component of the

TEM streamfunction except in the extratropics, the errors they introduce are apparently vanishingly small. This result instead points to temperature nudging (Fig. 4c) directly invigorating the Eulerian mean part of the circulation (Fig. 5; and see also Miyazaki et al. (2005); Akiyoshi et al. (2016)).

## 7   Errors in ozone and ozone transport

The errors in the dynamics project strongly onto errors in ozone (Fig. 13, 14) and carbon monoxide (Fig. 15, 16). To estimate

transport errors, we simply sum the eddy and residual circulation flux convergences on the right hand side of Eq. 11 and apply the same regression technique as before. However, to more directly assess the impact of each type of transport error on ozone and carbon monoxide, we multiply the error in the combined TEM residual flux and TEM eddy flux convergence by the e-folding timescale of the corresponding tracer. The timescales are estimated by determining the lag at which each tracer's autocorrelation drops to $1/e$ in the reference run. The e-folding timescale for ozone ranges from 5 days in the tropical upper

troposphere and lower stratosphere to 70 days in the extratropical stratosphere, while for carbon monoxide it ranges from 12 days in the tropical upper troposphere and lower stratosphere to 70 days in the extratropical stratosphere. Both of these timescales have substantial vertical structure, and in the troposphere are generally 10-20% shorter in the Northern Hemisphere.

With decreasing meteorology frequency, temporal errors in ozone peak at greater than 1% of the climatology in the upper troposphere/lower stratosphere and in the polar stratosphere (Fig. 13a). This pattern is mirrored by the increase in temporal

errors in the ozone flux convergence with decreasing meteorology frequency (Fig. 13b), which themselves reflect the temporal error in the EP flux divergence (Fig. 11b). Locally, at least, the temporal errors in ozone itself are consistent with errors in eddy mixing. However, there is a strong signal in the temporal error in the ozone flux, but not convergence, in the tropical stratosphere, implicating the deep branch of the residual circulation. This suggests that the residual circulation acts to accumulate errors in ozone along the equator-to-pole transport pathway downstream of photochemical ozone production, at which point

errors in the eddies become dominant.

The spatial error associated with nudging timescale in ozone peaks at up to 0.5% in the upper troposphere/lower stratosphere (Fig. 13c). This hemispherically-asymmetric pattern is somewhat consistent with spatial errors in ozone transport (Fig. 13d). In general, the anomalously high ozone in one hemisphere is associated with anomalous downward and equatorward transport from the middle stratosphere, while it is the opposite in the other hemisphere. In the Northern Hemisphere upper stratosphere,





erroneously lower ozone is associated with enhanced downward transport of ozone-poor air from the mesosphere, while in the Southern Hemisphere polar lower stratosphere, anomalously high ozone is associated with enhanced poleward and downward transport.

The mean error in ozone is not just limited to the tropics (Fig. 9), as decreasing nudging timescale leads to a reduction in ozone throughout the troposphere (Fig. 13e). This is associated with weakened vertical transport in the troposphere relative

to the climatology (Fig. 13f), and greater negative transport tendencies throughout the lower stratosphere as the anomalously ozone-poor air is spread around. However, there is a region of enhanced downward transport in the Southern Hemisphere that seems to be associated with the invigoration of the shallow branch of the TEM residual streamfunction with decreasing nudging timescale (Fig. 11e). Cancellation between this midlatitude blob of erroneously high ozone and the erroneously low ozone lofted through the tropical tropopause layer likely leads to the lack of robustness in the relationship between nudging

timescale and stratospheric ozone mean error (Fig. 8c). Ozone transport error is greater in magnitude and more complicated in structure than the errors in ozone itself, suggesting that chemistry may considerably damp transport error.

We can quantify the sources of these errors in transport by using the same "swapping" technique as before. Here, we will swap in the reference zonal mean terms, the eddy terms, and the zonal mean tracer field (separate from all other zonal mean terms) and recalculate the transport terms, the regression, and the conversion of the regression into an ozone impact using

the ozone e-folding timescale. The temporal error in transport is overwhelmingly due to eddy transport, which drives a global average 10% error in ozone (Fig. 14a). Spatial and mean error due to transport is more balanced between residual and eddy mixing at 2-6% error. Temporal and mean errors in transport are only substantially reduced when using the reference eddy flux. Because of the strong covariance between the temporal error in ozone transport and the temporal error in the EP flux divergence (Fig. 13b, Fig. 11b), we can infer this is primarily due to errors in the eddy circulation itself. No diagnostic swap

reduces the errors in residual circulation transport, demonstrating that the residual circulation acts to accumulate errors along transport pathways, rather than producing them locally.

## 8   Errors in carbon monoxide and carbon monoxide transport

Errors in carbon monoxide (Fig. 15) bear some resemblance to errors in ozone (Fig. 13). Carbon monoxide temporal error increases everywhere with decreasing meteorology frequency (Fig. 15a), with large errors in the upper troposphere/lower

stratosphere and in the polar stratosphere, reflecting the behavior of temporal errors in ozone (Fig. 15b). However, the errors in carbon monoxide and the carbon monoxide flux convergence are more peaked in the upper stratosphere just below the stratopause. This upper stratospheric temporal error is driven by upward and poleward transport in the tropics characteristic of the residual circulation, and downward and equatorward transport at the poles indicative of both eddy mixing and residual circulation transport (Fig. 15a,b, Fig. 11b). In the upper troposphere/lower stratosphere, the error is generally driven by downward

and poleward transport.

Carbon dioxide spatial error associated with nudging timescale is characterized by a hemispherically-asymmetric pattern throughout the whole atmosphere on the order of 0.5-1% of the climatology (Fig. 15c), consistent with erroneous cross-




hemispheric transport in the upper troposphere/lower stratosphere (Fig. 15d). Excess carbon monoxide over the pole (Fig. 15c) is fluxed from the pole to the midlatitudes (Fig. 15d), and vice versa in the other hemisphere. These patterns are out-of-phase

with the carbon monoxide errors. Either the transport errors follow from the carbon monoxide errors, or chemistry damps the transport errors.

Carbon monoxide increases everywhere in the troposphere and lower stratosphere when nudging timescale is decreased (Fig. 15e), consistent with erroneously enhanced upward transport of carbon monoxide into the lower stratosphere and erroneously weakened upward transport in the middle troposphere (Fig. 15f).

As with ozone, the errors in carbon monoxide due to transport are generally dominated by the eddy component of the carbon monoxide flux convergence (Fig. 16). Use of the reference eddy field results in substantially reduced temporal error due to the eddy carbon monoxide flux convergence, similar to ozone. However, there is a moderate reduction in spatial error due to the residual circulation flux convergence when the reference eddy field is used, and likewise a small reduction in spatial error due to the eddy flux convergence when the reference zonal mean is used (Fig. 16b). The former implicates erroneous carbon

monoxide transport associated with the adiabatic eddy recirculations, while the latter implicates regions with highly variable static stability or zonal wind, such as the upper troposphere/lower stratosphere and the stratopause. There is some indication that the mean error due to the residual circulation is reduced when using the reference tracer field, implicating transport through the upper troposphere-lower stratosphere where the mean error in carbon monoxide has its greatest impact on vertical gradients (Fig. 16c).

## 9   Conclusions and Discussion

Through an analysis of 18 CESM2 (WACCM6) specified dynamics simulations over the course of one simulated year, we have found that:

1. Meteorology frequency is the primary contributor to spatial and temporal errors in the tropospheric and stratospheric circulation

(a)  As meteorology frequency increases, spatial and temporal error decreases

   (b)  At a given meteorology frequency, nudging timescales shorter than 24-48 hours produce the lowest spatial and temporal error

2. Meteorology frequency is the primary contributor to temporal error in ozone and carbon monoxide, and their transports; while nudging timescale is the primary contributor to spatial error in ozone and carbon monoxide, and their transports

(a)  As meteorology frequency increases, temporal error decreases

   (b)  As nudging timescale increases, spatial error generally decreases, but reaches a minimum at a 12-24 hour nudging timescale





3.  Nudging timescale is the primary contributor to mean error in all fields, with the lowest mean error at 24-48 hour nudging timescales

Taken together, these results suggest that the maximum meteorology frequency possible, with a moderate nudging timescale of 12-24 hours, is an optimal configuration for CESM2 (WACCM6) 110-level specified dynamics simulations that balances the three different types of error across all of these fields.

Errors in tracers are generally the lowest in emission/production regions and highest at the end of characteristic transport pathways (Fig. 17). Convection and the tropospheric residual circulation create errors in ozone and carbon monoxide and

accumulate them in the upper troposphere/lower stratosphere through rapid overturning, like a conveyer belt. These errors propagate upward into the stratosphere via the residual circulation and get mixed horizontally by Rossby waves along the tropopause. Above this level, these errors are damped by photochemistry. Similarly, the deep branch of the residual circulation creates errors in ozone downstream of photochemical production in the tropical stratosphere and accumulates them in the polar stratosphere, where they are redistributed and accentuated by errors in Rossby wave transport. Because the dynamics in

the mesosphere cannot be reliably constrained without substantial instabilities, the photochemical production and downward transport of carbon monoxide through the mesosphere and into the upper stratosphere by the residual circulation (Minschwaner et al., 2010; Garcia et al., 2014) results in substantial errors in polar stratospheric carbon monoxide, which are mixed toward the equator by Rossby waves.

As this model configuration is computationally expensive, our analysis only spans 1 year due to the practical need to sweep

enough of the phase space of nudging parameters. We therefore believe that while the errors in the circulation are probably close to the value we would infer from longer simulations, the errors in the tracer fields should be seen as an underestimate, especially in the middle atmosphere. Circulation errors integrated over at least the stratospheric age of air timescale could lead to sustained increases in tracer error. It also may be the case that production and loss processes are strong enough to damp this increase. As the transport errors imply a greater error in ozone and carbon monoxide than is ever realized, photochemistry acts

as a major constraint on transport errors.

We can speculate how these results generalize to cases where the reference meteorology is produced by a different modeling system. Differences in cloud parameterizations and convective transport might lead to slightly different optimal nudging timescales for tracer transport. Recall that if a parameterization such as convection adjusts temperatures in the reference meteorology, temperature nudging will essentially do the work of the convective parameterization if the nudging timescale is short

enough and lead to errors in the specified dynamics simulation's clouds (Fig. 4). It seems unlikely that the timescale of processes like convection will be so different between modeling systems so as to change these results, though. Instead, we expect differences in the climatology between the reference meteorology and specified dynamics simulation to present the greater challenge.

The relatively strong dependence of errors in chemical tracers and clouds on both nudging timescale and meteorology

frequency is especially concerning because specified dynamics simulations, by their very nature, are intended to constrain the circulation and isolate its role in chemical weather and climate. Few of the tracer errors can be tied to particular and





local processes, highlighting the degree to which the circulation integrates errors. However, the high sensitivity of these errors suggests that any modeling interventions could have major impacts.

It is worth asking sjust how much we can reduce the errors in specified dynamics schemes based on the linear relaxation of winds and temperatures. Nudging potential vorticity, rather than the individual components of the wind, may better constrain the divergent circulation (Li et al. , 1998; Pulido, 2014) and curtail erroneous circulation responses (DeWeaver and Nigam, 1997). However, potential vorticity cannot constrain the absolute temperature, which is critical for temperature-dependent chemistry.

If the relative magnitude of the nudging tendencies at a given timestep can be used as a proxy for the instability of the current atmospheric state, analogous to bred vectors (Cai et al., 2003), our current constant-timescale linear relaxation could be easily transformed into a flow-adaptive timescale. The flow-adaptive timescale could be tuned to nudge the meteorology at a fast timescale only when absolutely necessary, as meteorology errors are relatively insensitive to nudging timescales faster than 24 hours. It could also be made aware of active convection and either reduce the nudging timescale accordingly, or switch off temperature nudging to allow the convective physics the freedom to modify the temperature field and transport constituents. Adaptive linear nudging of potential vorticity and temperature may provide a tractable solution for further reducing errors in tracers and clouds, while still sufficiently constraining the circulation.

*Code and data availability.* Instructions for acquiring CESM source code can be found at https://www.cesm.ucar.edu/, with a user guide hosted at https://ncar.github.io/CAM/doc/build/html/users_guide/index.html. Source code modifications are hosted at Zenodo for the online TEM tracer flux diagnostics (Davis, 2021a). Please note these were developed for CESM version 2.0, and may not work for other releases of the model. Model output necessary to replicate these results is hosted at Zenodo (Davis, 2021b).

*Author contributions.* S. Tilmes conceived of the project and set up the model configuration, P. Callaghan modified the physics order source code and wrote the nudging scheme description in the manuscript, N. A. Davis created the online tracer output source code, set up and ran the model experiments, performed the analysis, and wrote the manuscript, and S. Tilmes, P. Callaghan, and I. R. Simpson revised the manuscript.

*Competing interests.* The authors declare no competing interests are present.

*Acknowledgements.* The CESM project is supported primarily by the National Science Foundation (NSF). This material is based upon work supported by the National Center for Atmospheric Research, which is a major facility sponsored by the NSF under Cooperative Agreement No. 1852977. Computing and data storage resources, including the Cheyenne supercomputer (doi:10.5065/D6RX99HX), were provided by the Computational and Information Systems Laboratory (CISL) at NCAR. We thank all the scientists, software engineers, and administrators who contributed to the development of CESM2.



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





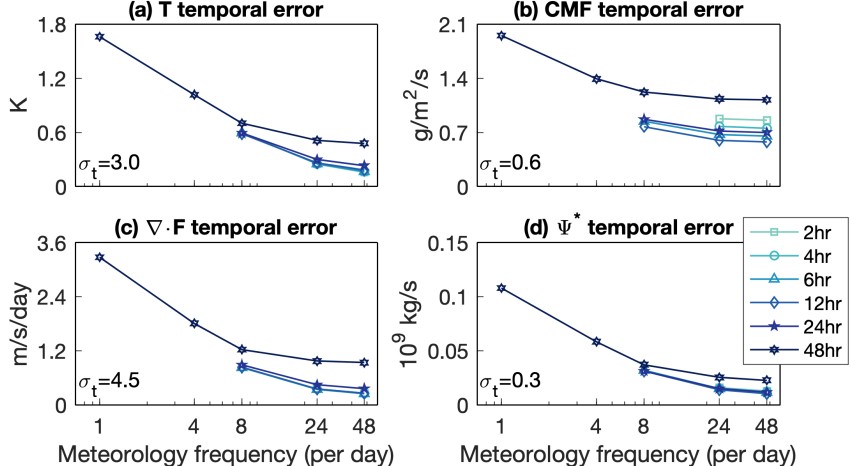

**Figure 1.** Globally- and vertically-averaged root-mean-square temporal error in (a) temperature, (b) convective mass flux, (c) EP flux divergence, and (d) the TEM streamfunction as a function of meteorology frequency (horizontal axis) and nudging timescale (see legend). For reference, the globally- and vertically-averaged temporal standard deviation of each field is shown in each panel.

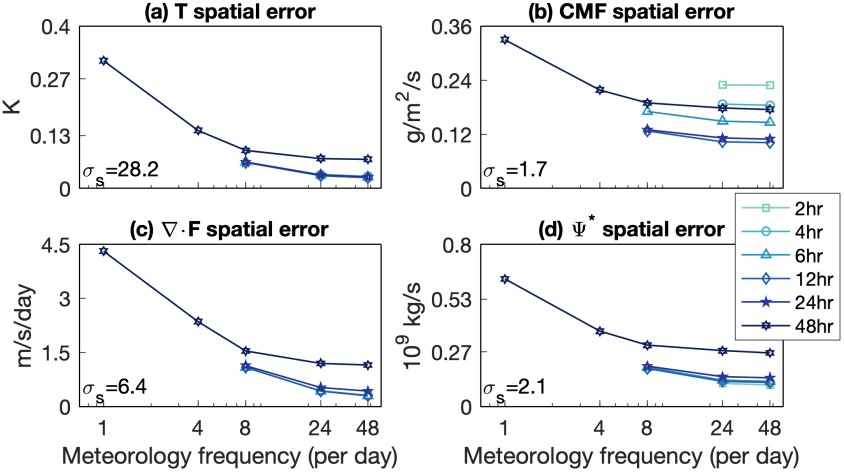

**Figure 2.** As in Fig. 1, but for the root-mean-square spatial error, and the globally- and vertically-averaged spatial standard deviation of each field is shown in each panel.



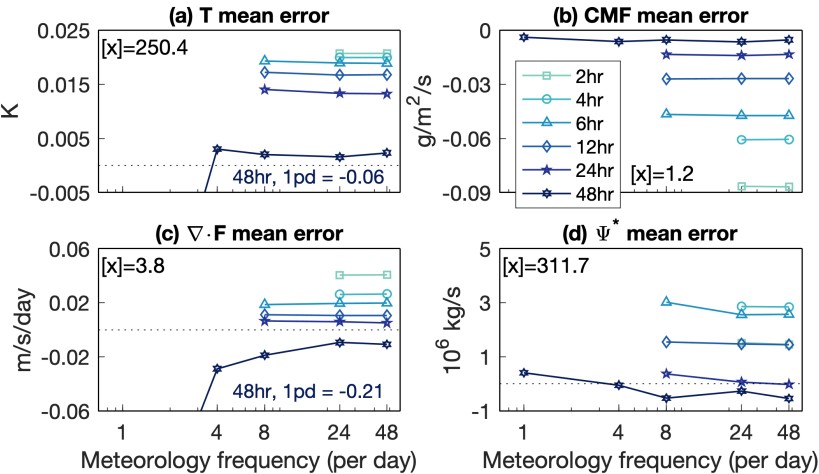

**Figure 3.** As in Fig. 1, but for the mean error, and the globally- and vertically-averaged absolute value of each field is shown in each panel.

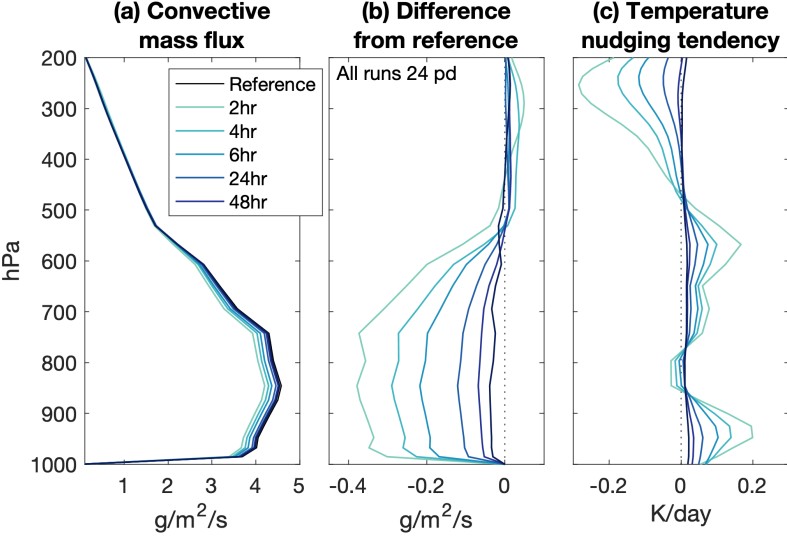

**Figure 4.** Tropical convective mass flux (a) mean and (b) difference from the reference meteorology, and (c) temperature nudging tendency for all nudging timescales at 24 meteorology updates per day. Average taken from 20S to 20N.





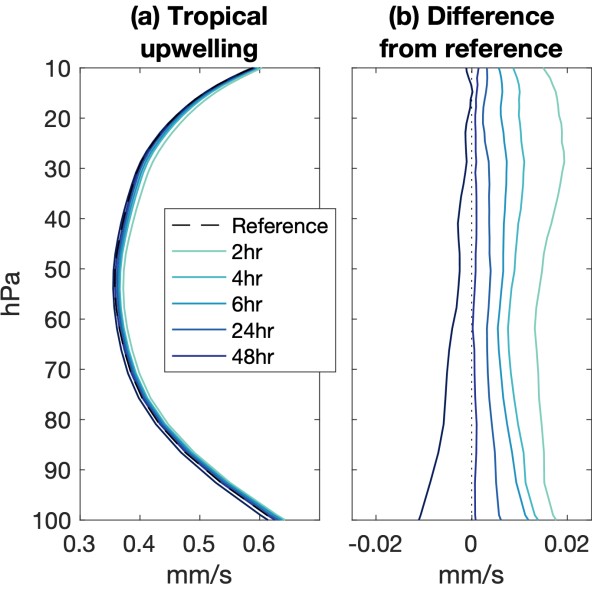

**Figure 5.** Tropical stratospheric upwelling (a) mean and (b) difference from the reference meteorology for all nudging timescales at 24 meteorology updates per day. Tropical stratospheric upwelling is defined as the area-average of all annual mean upwelling vertical velocities at each vertical level.

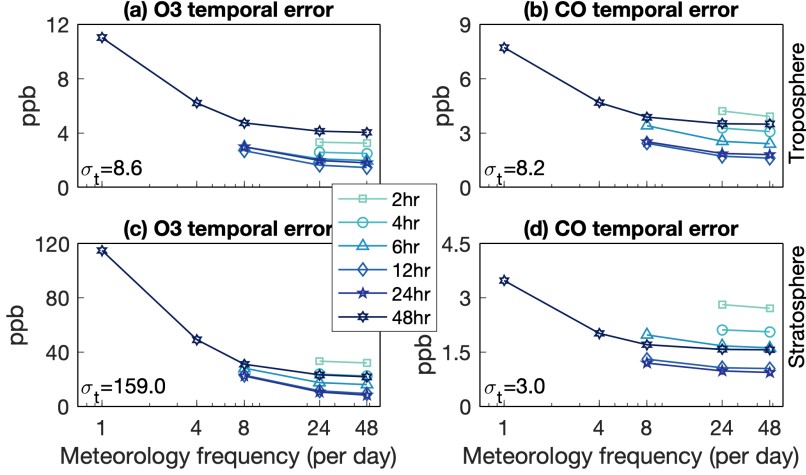

**Figure 6.** Globally- and vertically-averaged root-mean-square temporal error in tropospheric (a) ozone and (b) carbon monoxide and stratospheric (c) ozone and (d) carbon monoxide. For reference, the globally- and vertically-averaged temporal standard deviation of each field is shown in each panel.

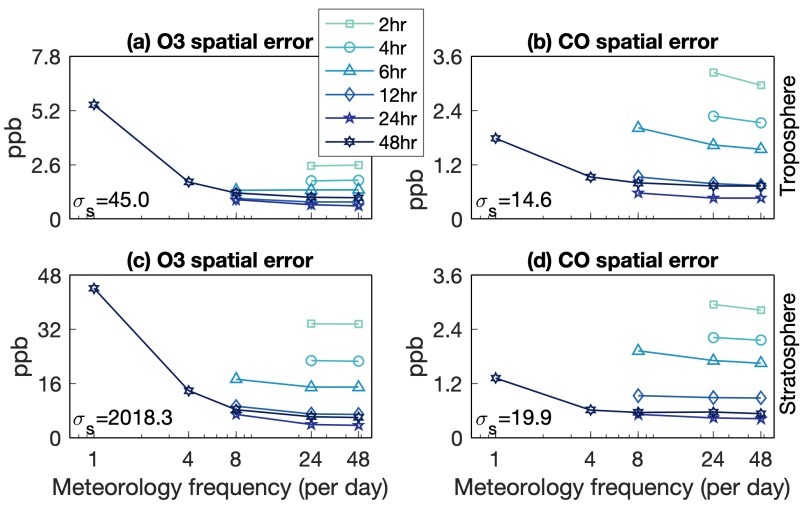

**Figure 7.** As in Fig. 6, for the root-mean-square spatial error, and the globally- and vertically-averaged spatial standard deviation of each field is shown in each panel.

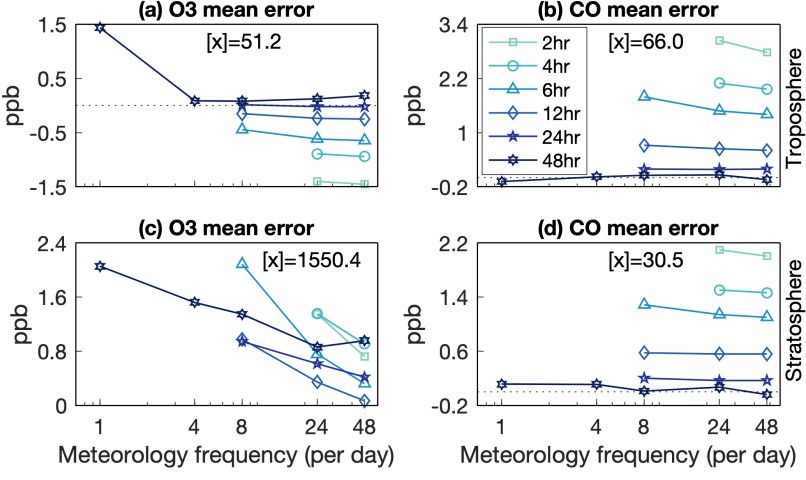

**Figure 8.** As in Fig. 6, but for the mean error, and the globally- and vertically-averaged absolute value of each field is shown in each panel.





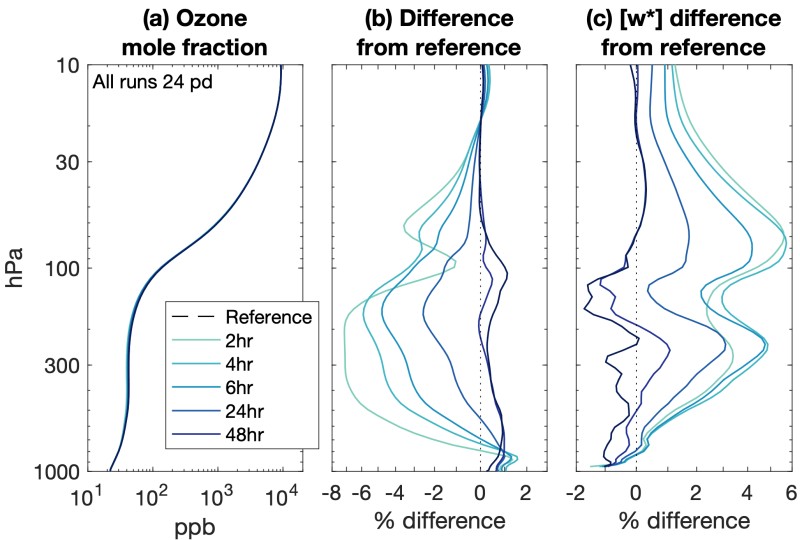

**Figure 9.** Tropical ozone (a) mean and (b) percent difference from the reference meteorology, and (c) residual vertical velocity for all nudging timescales at 24 meteorology updates per day. Average taken from 20S to 20N.

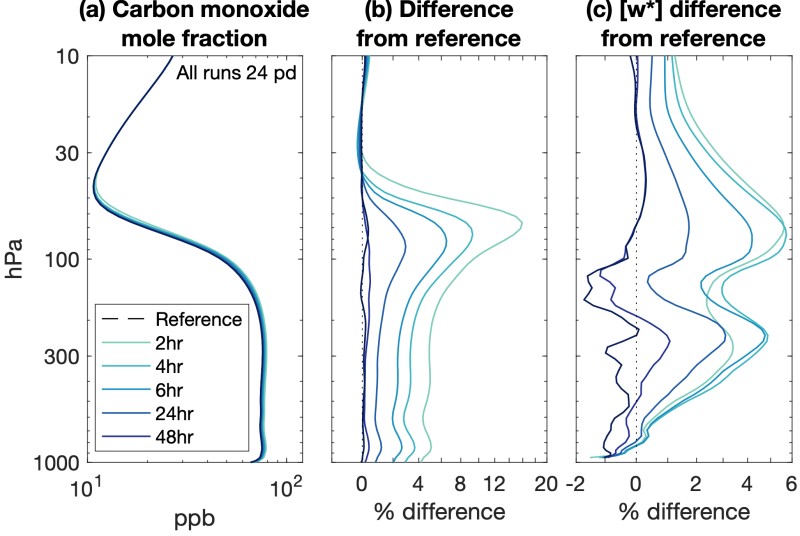

**Figure 10.** As in Fig. 9, but for carbon monoxide.

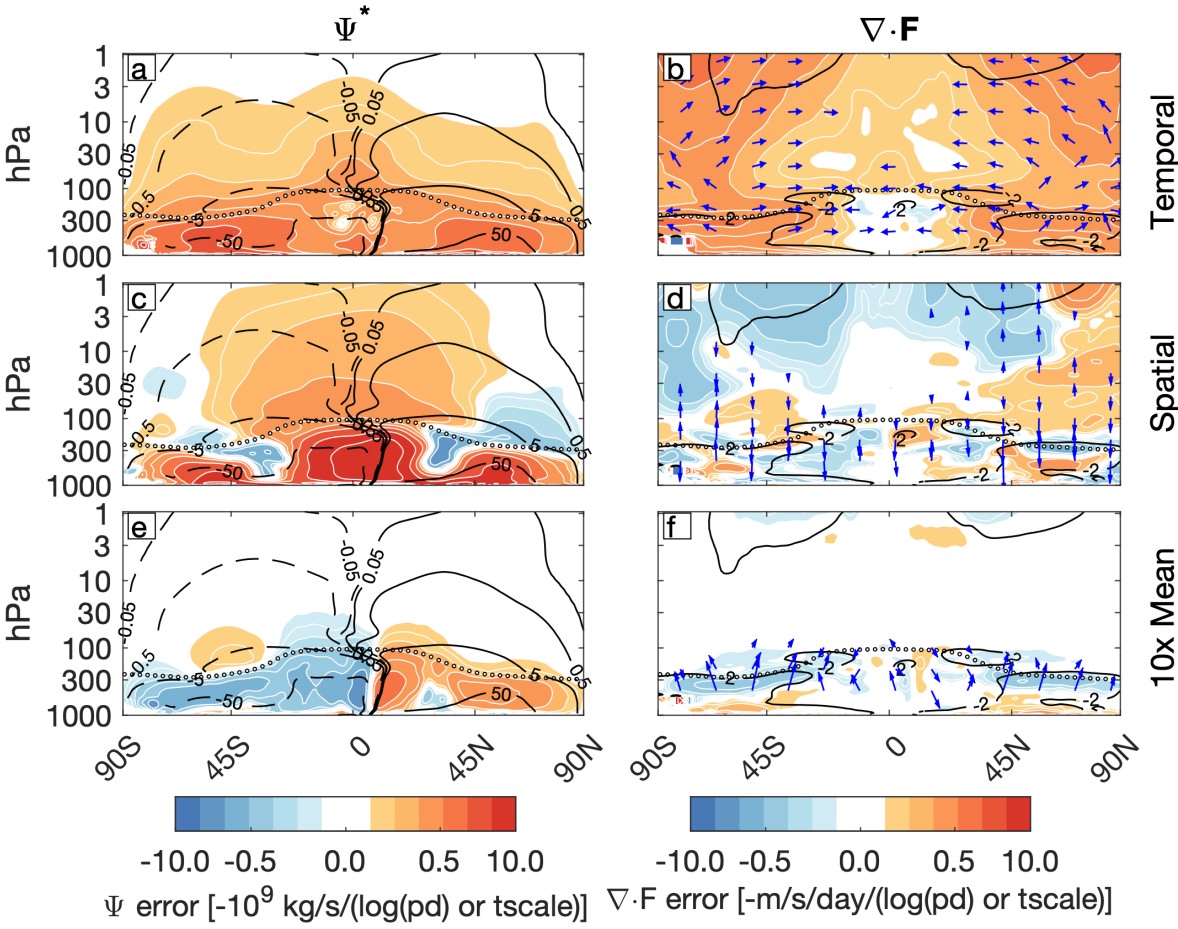

**Figure 11.** Negative of the projection of (top row) root-mean-square temporal error, (middle row) root-mean-square spatial error, and (bottom row) mean error in the (left column) TEM streamfunction and (right column) EP flux and divergence onto (top and middle rows) logarithm of meteorology frequency and (bottom row) nudging timescale. Projection in shading (logarithmic scale), climatology in contours, and EP flux in vectors, scaled as in Edmon et al. (1980). In (a), (c), and (e) the climatological TEM streamfunction is contoured 0.05, 0.5, 5, and $50 \times 10^9$ kg/s, with positive values solid and negative values dashed. In (b), (d), and (f) the climatological EP flux divergence is contoured every 2 m/s/day, with negative values solid. The tropopause is shown by the black and white dotted contour.

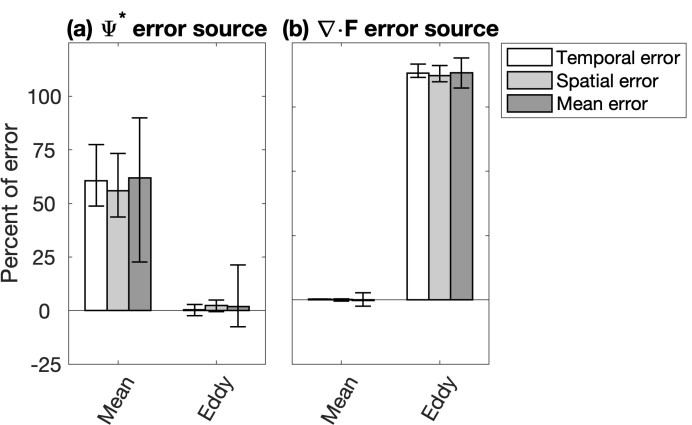

**Figure 12.** Percent of the temporal, spatial, and mean error in the (a) TEM streamfunction and (b) EP flux divergence attributable to the zonal mean or eddy fields. Whiskers indicate the maximum and minimum across all nudging simulations.



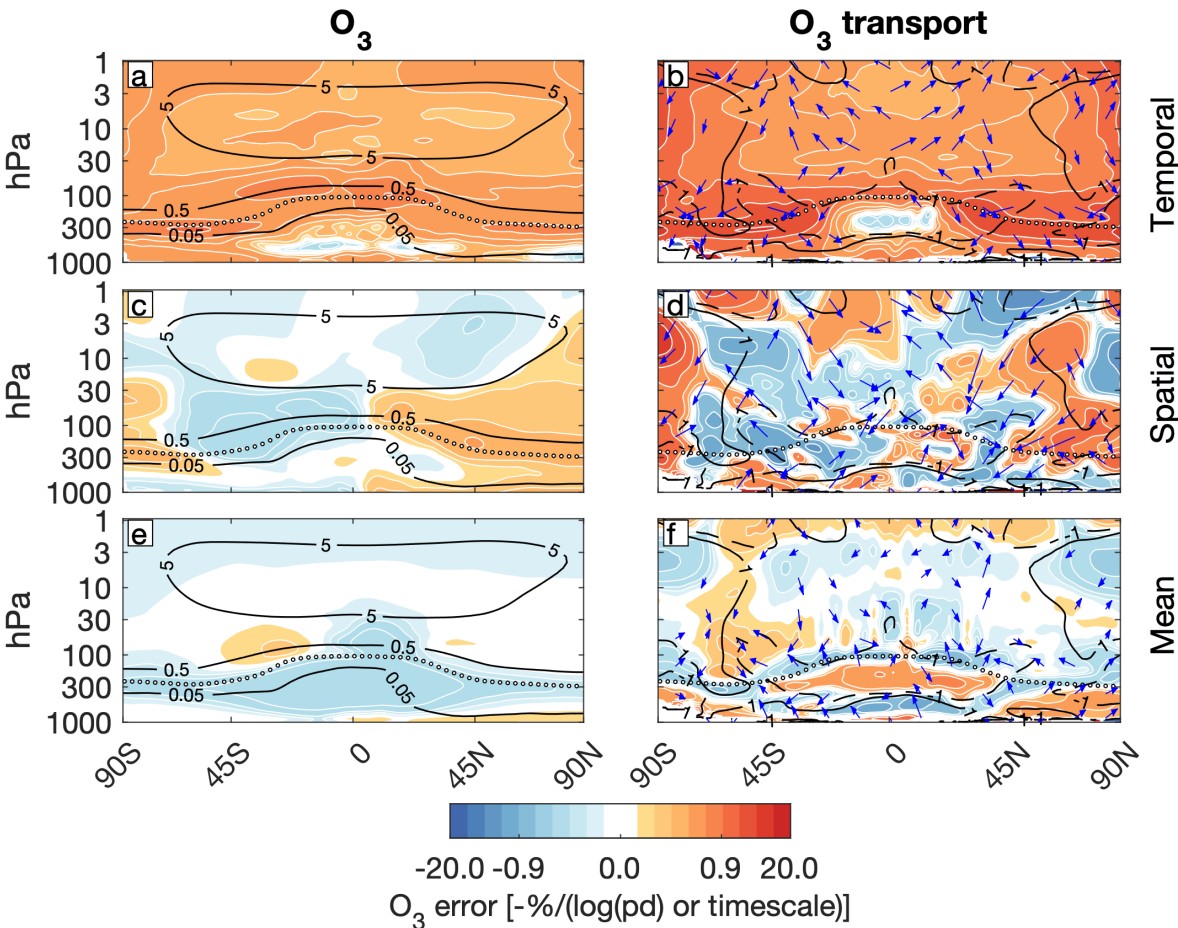

**Figure 13.** Negative of the projection of (top row) root-mean-square temporal error, (middle row) root-mean-square spatial error, and (bottom row) mean error in (left column, shading) the ozone mole fraction and (right column, vectors) the combined TEM residual circulation and eddy ozone flux and (shading) convergence onto (top row) logarithm of meteorology frequency and (bottom row) nudging timescale. Climatology in contours in (left column) parts-per-thousand and (right column) %/day, with negative values dashed. The tropopause is shown by the black and white dotted contour.



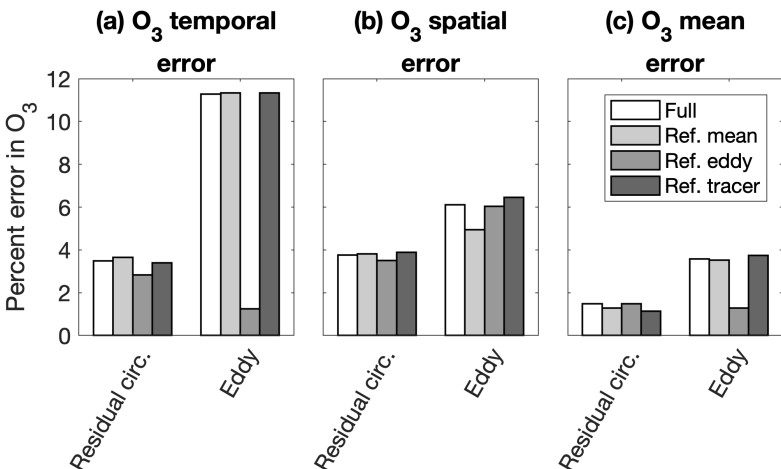

**Figure 14.** Global average (a) root-mean-square temporal error, (b) root-mean-square spatial error, and (c) sign-adaptive mean error in ozone attributable to the TEM residual circulation and TEM eddy flux convergences, based on the projections in the right column of Fig. 13

.

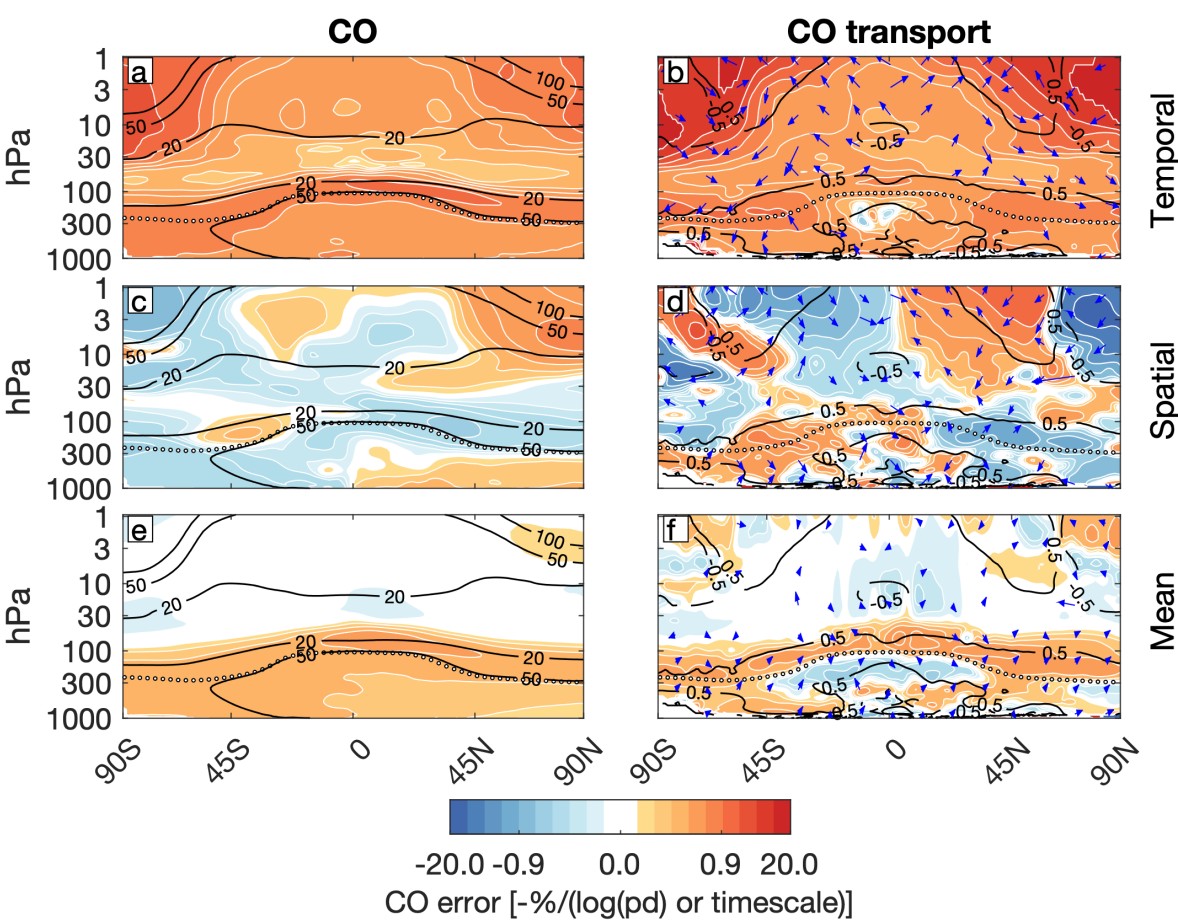

**Figure 15.** As in Fig. 13, but for carbon monoxide.

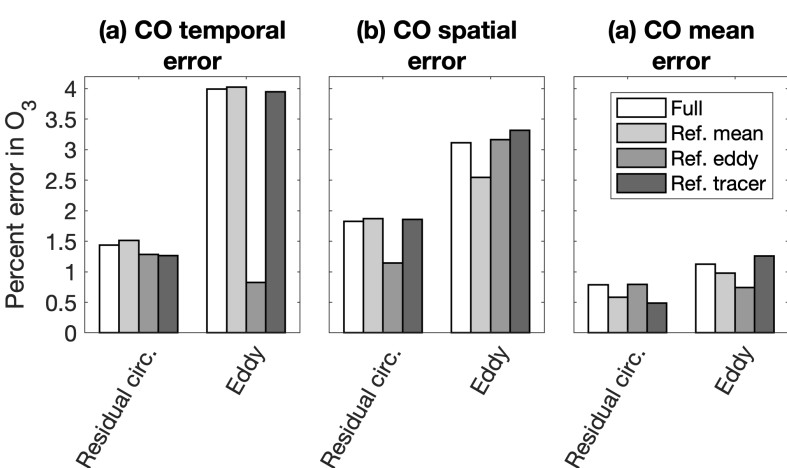

**Figure 16.** As in Fig. 14, but for carbon monoxide, and based on the projections in the right column of Fig. 15.





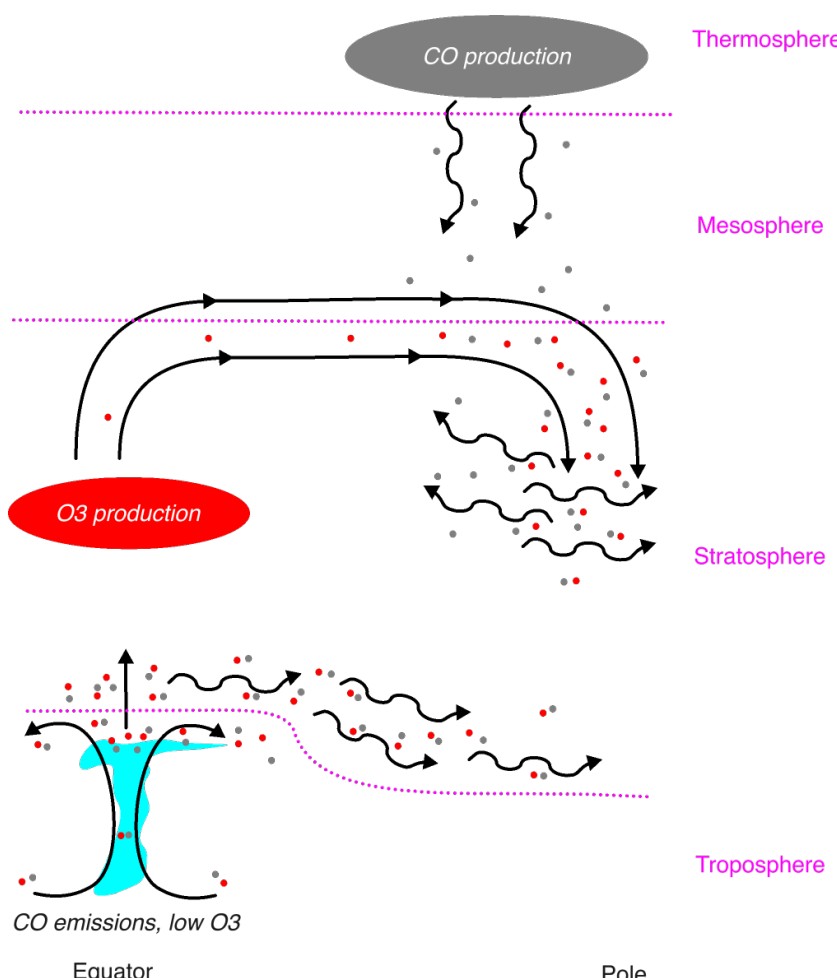

**Figure 17.** Schematic of transport errors produced by the specified dynamics scheme. The residual circulation is illustrated by the steady black arrows, while the eddies are illustrated by the squiggly black arrows. Convection is shown by the light blue cloud, while photochemical production regions are shown by the large bubbles for (red) ozone and (grey) carbon monoxide, with the concentration indicating the severity of the error. Errors are shown by the small dots for (red) ozone and (grey) carbon monoxide. Dotted lines delineate the tropopause and stratopause.