# Peer review of "Specified dynamics scheme impacts on wave-mean flow dynamics, convection, and tracer transport in CESM2 (WACCM6)"

_Atmospheric Chemistry and Physics, 2021_

## Referee Comment (RC1)

**Overall comments**

In this study the authors explore the impacts of various combinations of specified-dynamics (SD) schemes and its impacts on convective and wave-mean flow dynamics and the associated tracer transport in the most recent version of the high top CESM2-WACCM version. The novelty of this study lies in providing a thorough test of the specified dynamics scheme by nudging the model circulation to its own free-running output, unlike previous studies in the literature where the model was nudged towards a reanalysis meteorology. This is indeed a very important topic as recent studies have highlighted that specified dynamics schemes, as implemented in various model frameworks, do not constrain the stratospheric mean meridional circulation or the underlying trends in the reanalysis products, ultimately adversely affecting transport processes. The study provides an in-depth analysis of the effects of combining different parameters including the meteorology frequency and the nudging timescale within the specified dynamics scheme; all by nudging the zonal mean winds and the temperature (u,v and T). The authors analysed these parameters with 18 different experiments that span one year with WACCM to understand how nudging affects the error structure and propagation in the dynamics and the transport of species in the troposphere and the stratosphere. The study points out several deficiencies in the current application of the nudging scheme in WACCM which may be generalised for other models as well. Moreover, one of the main findings is that even for just one model year, the residual circulation and eddy mixing processes accumulate tracer errors at the end of their characteristic transport pathways in the upper troposphere/lower stratosphere as well as in the polar stratosphere. However, the authors identify nudging parameter combination scenarios within WACCM's configuration that often minimise errors in the circulation and tracer transport. Overall, this study is a cautionary tale of the degree of impact of nudging schemes on various aspects of the coupled chemistry-climate system identifying the artificial limitations and implications that arise from the use of such a scheme and it greatly improves our understanding of such processes. It fits naturally within the scope of Atmospheric Chemistry and Physics journal and I recommend this study to be accepted and published with minor revisions. I don't have any major reservations related to specific parts of the text, but some minor specific comments and recommendations that may enhance the read of the paper follow just below.

**Specific comments**

**Paragraph starting at line 31**: I find that the CO discussion is short, and it could be enhanced a bit more in this paragraph.

**Page 3, line 63**: Please specify which meteorological products were used in Kinnison et al. (2007) or at least state that one of them was from an older version of WACCM.

**Page 3, line 65**: Specify that these results are based on the Chemistry-Climate Model Initiative (CCMI) output and provide the time scales (multi-decadal, climatological) in which the SD model spread in those studies was found to be as large or larger than in the free-running (FR) simulations. Overall, I feel you can mention some of the most relevant details of these multi-modelling studies that first investigated the impacts of the implementation of SD schemes in a multi-modelling framework. I would also add the Orbe et al., (2020) paper as a reference.

**Page 3, lines 82-83**: Apart from $O_3$ being the obvious choice for looking at transport processes especially in the stratosphere, what was the defining criteria you selected CO in the study and

not another tracer? The reader could find a short comment around the choice of the tracers interesting.

**Page 4, lines 96-98:** Perhaps I'm not familiar enough with this but would you expect the met list nudging implementation to produce different results to the ones you are showing with the "alternative" dynamical-core-independent nudging technique?

**Page 4, lines 107-108**: Please clarify that these refer to the the absolute values.

**Page 4, lines 132-136**: Although the text is quite clear, perhaps a table with all the combinations including the ones that require an increase in advection sub-cycling crossed out might be helpful to the reader.

**Page 5, lines 142-144**: Very interesting indeed. Perhaps you could at least make a note of this (or even caveat) in the discussion section of the document.

**Page 8, lines 213-214:** How do you estimate the percentage change relative to the total variability? Is it ±3.x stddev assuming a normal distribution? Please clarify in the text or perhaps add a short note of this in the methodology section.

**Page 8, start of line 217**: Please clarify here that longer timescales refer to the nudging timescale.

**Page 8, end of line 232**: clarify that in this case the minimal global mean error occurs at the longer timescale.

**Page 8, lines 235-236**: However, they do not seem to scale. In Figure 3d the 2hr line is also missing. Is it out of bounds in this plot?

**Page 8, lines 238-239**: This is more of a remark than something to address; you can add a note of this perhaps. It's quite interesting to see in Figure 5b that the 24h nudging performs very well throughout the depth of the stratosphere.

**Page 9, start of line 264**: You could maybe combine Figures 9 and 10 in a single figure with 5 panels? Currently Figure 9c and 10c are duplicates.

**Page 10, lines 278-279**: This is unclear to me. Do you mean that you will focus on the mean error negative regression values or the regression values with decreasing meteorology frequency and/or nudging timescale?

**Page 10, line 300**: Are these EP flux vector anomalies wrt to the reference meteorology or do they represent the spatial errors? It seems the latter but please clarify that in the caption of Fig. 11 or in a relevant part of the text.

**Page 10, lines 307-308:** The way you have written the brackets denoting Fig. 3c and d prompt the reader to find the tropospheric and stratospheric Psi and EPF Div mean errors. This seems to be not the case as the quantities shown are the global mean vertically averaged mean errors. Please rephrase this sentence so it conveys what you meant originally. "Global average mean errors in the circulation (Fig. 3c,d)... in the stratosphere."

**Page 11, lines 329-331**: Perhaps a lat/height cross section figure of the e-folding $O_3$ and CO timescales could be added in the supplement. I wouldn't consider that a must, but it might be helpful to the reader. I don't feel that this is necessary to be shown in the paper; you could just include it in the responses.

**Page 11, end of line 332**: "…Northern Hemisphere" – compared to?

**Page 12, line 362**: "…are only substantially…" – why only?

**Page 13, line 378**: Please note that you still talk about the UTLS as in the mid to upper stratosphere it seems to me that the fluxes do not have the same structure. Also "…over the pole" – clarify which pole?

**Page 13, line 382**: Clarify compared to what CO is increased? Reference climatology or that it is the mean error that increases?

**Page 14, line 427**: Please provide an average estimated range of the stratospheric AoA for the reader.

**Typos**

**Page 3, line 60:** "…one *of* none…" – correct to or

**Page 3, line 72**: "strength" *of the* "meridional circulation,"

**Page 15, line 444**: "asking sjust" – just

**References**

Orbe, C., Plummer, D. A., Waugh, D. W., Yang, H., Jöckel, P., Kinnison, D. E., Josse, B., Marecal, V., Deushi, M., Abraham, N. L., Archibald, A. T., Chipperfield, M. P., Dhomse, S., Feng, W. and Bekki, S.: Description and Evaluation of the specified-dynamics experiment in the Chemistry-Climate Model Initiative, Atmospheric Chemistry and Physics, 20(6), 3809–3840, doi:10.5194/acp-20-3809-2020, 2020.

---

## Referee Comment (RC2)

**Review of "Specified dynamics scheme impacts on wave-mean flow dynamics, convection, and tracer transport in CESM2 (WACCM6)" by Davis et al.**

In this manuscript, entitled "Specified dynamics scheme impacts on wave-mean flow dynamics, convection, and tracer transport in CESM2 (WACCM6)", the authors present a rigorous examination of the sensitivities of various aspects of the transport and dynamical circulation in CESM2 to nudging the large-scale flow.  A systematic exploration of how errors depend on both meteorology frequency as well as nudging timescale is presented.  As this manuscript presents an unparalleled level of information, detailing the behavior of the nudging framework within CESM2, it represents an important study that is certainly deserving of publication. I commend the authors for performing this type of evaluation, which is somewhat lacking in the literature and, as illustrated by the authors, riddled with complexities and nuances.  There are, however, a few ways in which the manuscript can be improved to both clarify the applicability of these results to other (namely reanalysis) fields as well as to focus the discussion on the major findings in order to communicate more clearly the first order results. As such, my recommendation is "major revisions", but I emphasize to the authors that this study has strong potential and will constitute a valuable contribution to the field, provided that these revisions are incorporated.

**Major Comment 1:**

While I agree with the authors that using meteorology produced from the same underlying GCM is, indeed, the cleanest way to assess the performance of the nudging scheme, one wonders how the major conclusions from the study change (if they do) upon nudging to reanalysis fields.  In particular, one of the central challenges with nudging is striking the "correct" balance between sufficiently constraining to the observed (reanalysis) fields, while also not doing too much harm in the actual act of nudging (i.e. producing dynamical inconsistencies in the flow that may further degrade the transport characteristics of the simulation). Therefore, while it is certainly worthwhile to evaluate the performance of the nudging scheme within the (parameter and flow) space of the parent GCM, ultimately what matters in the end is how that scheme operates in the combined "GCM-reanalysis" space.   In other words, one needs information (not provided in the current manuscript) about the underlying meteorological biases in the underlying GCM (CESM2). How confident are the authors that a scheme that reproduces CESM2 fields also reproduces other (reanalysis) fields, given what might be quite large model biases?   The authors have shown that the behavior of the nudging scheme does not necessarily exhibit nice convergence properties (for example — the minimization of the convective mass flux errors at a 12 hour nudging timescale, with increased errors at both shorter and longer nudging timescales (lines 220-222) ).  If the nudging scheme presents such complicated behavior, when constrained with its own fields, one can imagine the behavior might become still more complex when applied to reanalysis fields, potentially leading to major differences in the author' conclusions.

To this end, I think a necessary addition to the manuscript is the addition of a few simulations wherein CESM2 is nudged to MERRA-2 (or any reanalysis product of the authors' choosing) using a subset of the nudging timescale parameter combinations that are explored in the original set of experiments.  I understand that these caveats are mentioned in lines 431-438 but I am not

convinced that this is sufficient to address this issue. I am not suggesting an exhaustive set of runs but, rather, two or three simulations that demonstrate that the main findings of the study also hold when nudging to reanalysis fields.  Since CESM2-SD is already set up to nudge to reanalysis fields, I cannot imagine that this is an unreasonable ask.

**Major Comment 2:**

I commend the authors for performing a quite exhaustive examination of the errors in various circulation and transport diagnostics within the context of the simulations considered in this study.  At the same time, however, the complexity of the results renders the manuscript very descriptive, and it can be difficult for the reader to extract the key results from some of the more secondary points.  In particular, the separate discussions of the ozone and carbon monoxide errors (Sections 7 and 8) are quite long and nuanced.  Is there not a way to combine into a single section that is prefaced by a paragraph highlighting the common features among these two constituents (with respect to their response to nudging), followed by a discussion of each regarding the specific characteristics of each field?  At present, the number of details presented in the manuscript renders it a bit hard to follow and the authors should better emphasize throughout the main "take-away" messages (which are currently reserved only for the conclusions).

**Minor Comments:**

1. Line 155: Shouldn't the stratospheric vs. tropospheric averages be somewhat latitude-dependent, at least enough to distinguish between first-order differences in tropopause height? In particular, my concern is that in the extratropics 200 hPa is already well within the lower stratosphere and yet still quite tropospheric in the tropics.  Why not use something a bit more physically based? (i.e. 300 hPa for latitudes > 40S/N and 100 hPa for equatorward latitudes)?

2. Figure 4:  This is a very important figure, and it is nice to see this result documented so well.  It could be worth noting in the text when describing this result that similar behavior was observed in Figure 3b,d in the study indicated below, albeit for only two nudging timescales (and using CAM, nudged to MERRA). Consistent with the results presented in this study, that figure shows that the convective mass fluxes in the 5hr nudged simulation differed substantially more than the fluxes from the 50 hour nudged simulation, relative to MERRA. Perhaps it is worth highlighting this consistency between the two studies.

Orbe C., D. W. Waugh, H. Yang, J. F. Lamarque, D. Kinnison, and S. Tilmes (2016), Tropospheric Transport Differences Between Models Using the Same Large-Scale Meteorological Fields, *Geophysical Research Letters,* 44(2), 1068-1078.

3. Line 165: "Negative values of the SAME indicate that the field is weaker in magnitude than it is in the reference simulation - up to and including opposite in sign".  I do not necessarily agree with this description, and I think the issue is just one of unclear wording.  For example, consider

that x_ref=-2 and x_sd=3. Then, according to (5), SAME<0.  And, yet, the field is *greater* in (absolute) magnitude than it is in the reference simulation.  Perhaps the "up to an including opposite in sign" just needs to be clarified that it refers explicitly to x_sd.

4.  Line 401: Is conclusion 1(b) true?  Don't the errors in convective mass fluxes *increase* at nudging timescales shorter and longer than 12 hr?

---

## Author Comment (AC1)

**Overview of major revisions to the manuscript**

Thank you to the three anonymous reviewers for the constructive feedback. We have worked hard to address your concerns and would like to present a summary of major changes to the manuscript.

      1. Addition of three new experiments with WACCM nudged to GEOS meteorology from January 1, 2018 - December 31, 2018. We chose our "best case" configuration based on the results of the existing analysis, as well as two somewhat similar configurations we expected to produce higher errors. We have added a new figure, Figure 20, detailing these results. They largely reflect the results of the rest of the analysis, but with an important caveat that it is both the convective mass flux and the meridional circulation that are difficult to constrain relative to GEOS.

      2. Merging of the ozone and carbon monoxide sections. Rather than having two separate sections, with many repeated statements in the second section, we combined the ozone and carbon monoxide discussions so that we could directly compare and contrast their results. We believe this provides the reader with a better understanding of the robust impacts of nudging-induced circulation errors on these tracers.

      3. Additional discussion of the health and radiative impacts of carbon monoxide in the introduction.

      4. Fixed an error in Eq.'s 6 and 12. The correct formulation for the residual streamfunction flux terms in Eq.'s 6 and 12 involve a 90 degree rotation of the divergence operator through its cross product with the zonal unit vector.

**Responses to Reviewer #1**

*Overall comments*

*In this study the authors explore the impacts of various combinations of specified-dynamics (SD) schemes and its impacts on convective and wave-mean flow dynamics and the associated tracer transport in the most recent version of the high top CESM2-WACCM version. The novelty of this study lies in providing a thorough test of the specified dynamics scheme by nudging the model circulation to its own free-running output, unlike previous studies in the literature where the model was nudged towards a reanalysis meteorology. This is indeed a very important topic as recent studies have highlighted that specified dynamics schemes, as implemented in various model frameworks, do not constrain the stratospheric mean meridional circulation or the underlying trends in the reanalysis products, ultimately adversely affecting transport processes. The study provides an in-depth analysis of the effects of combining different parameters including the meteorology frequency and the nudging timescale within the specified dynamics scheme; all by nudging the zonal mean winds and the temperature (u,v and T).The authors analysed these parameters with 18 different experiments that span one year with WACCM to understand how nudging affects the error structure and propagation in the dynamics and the transport of species in the troposphere and the stratosphere. The study points out several deficiencies in the current application of the nudging scheme in WACCM which may be generalised for other models as well. Moreover, one of the main findings is that even for just one model year, the residual circulation and eddy mixing processes accumulate tracer errors at the end of their characteristic transport pathways in the upper troposphere/lower stratosphere as well as in the polar stratosphere. However, the authors identify nudging parameter combination scenarios within WACCM's configuration that often minimise errors in the circulation and tracer transport. Overall, this study is a cautionary tale of the degree of impact of nudging schemes on various aspects of the coupled chemistry-climate system identifying the artificial limitations and implications that arise from the use of such a scheme and it greatly improves our understanding of such processes. It fits naturally within the scope of Atmospheric Chemistry and Physics journal and I recommend this study to be accepted and published with minor revisions. I don't have any major reservations related to specific parts of the text, but some minor specific comments and recommendations that may enhance the read of the paper follow just below.*

Thank you for your review, we appreciate it.

*Specific comments*

*Paragraph starting at line 31: I find that the CO discussion is short, and it could be enhanced a bit more in this paragraph.*

Yes, we seem to have completely neglected CO. We've added a discussion on lines 38-41 concerning the health and (indirect via ozone) radiative effects of CO.

*Page 3, line 63: Please specify which meteorological products were used in Kinnison et al. (2007) or at least state that one of them was from an older version of WACCM.*

We now note the meteorology is from WACCM1b and from two ECMWF products.

*Page 3, line 65: Specify that these results are based on the Chemistry-Climate Model Initiative (CCMI) output and provide the time scales (multi-decadal, climatological) in which the SD model spread in those studies was found to be as large or larger than in the free-running (FR) simulations. Overall, I feel you can mention some of the most relevant details of these multi-modelling studies that first investigated the impacts of the implementation of SD schemes in a multi-modelling framework. I would also add the Orbe et al., (2020) paper as a reference.*

We agree some more details here are helpful - we've added Orbe et al. (2020), note the timescale, and in the proceeding lines now note the evidence that timescale improves the simulation of meteorology (Merryfield et al. 2013) but impacts convection (Orbe et al. 2017) ad the meridional circulation (Hardiman et al. 2017).

*Page 3, lines 82-83: Apart from O3 being the obvious choice for looking at transport processes especially in the stratosphere, what was the defining criteria you selected CO in the study and not another tracer? The reader could find a short comment around the choice of the tracers interesting.*

We've added a short discussion on lines 89-91 to motivate our choice. "These tracers have contrasting source and loss regions. Ozone is photochemically produced in the tropical stratosphere, while carbon monoxide is produced at the surface by combustion and photochemically produced in the mesosphere/lower thermosphere. Together they may provide a comprehensive sample of circulation impacts on tracer transport." The idea is that any common behavior indicates robust effects, more so than it would for one tracer, or for two tracers with similar production/loss regions.

*Page 9, start of line 264: You could maybe combine Figures 9 and 10 in a single figure with 5 panels? Currently Figure 9c and 10c are duplicates.*

We grant there is repetition in the third panels, but at the same time we prefer to maintain this aspect ratio in the panels. Our experience with typesetting is that this kind of plot would need to be rather large on the page to see any detail.

*Page 10, lines 278-279: This is unclear to me. Do you mean that you will focus on the mean error negative regression values or the regression values with decreasing meteorology frequency and/or nudging timescale?*

This sentence was unclear, we now clarify on line 307 that "We will display the negative of all error regressions - that is, for decreasing nudging timescale and decreasing meteorology frequency."

*Page 10, line 300: Are these EP flux vector anomalies wrt to the reference meteorology or do they represent the spatial errors? It seems the latter but please clarify that in the caption of Fig.11 or in a relevant part of the text.*

Yes, that is correct - the fluxes are the regression. We now clarify on line 304, "Therefore, we will focus on the regression of temporal and spatial errors in the EP flux and its divergence", as well as in the caption.

*Page 10, lines 307-308:The way you have written the brackets denoting Fig.3c and d prompt the reader to find the tropospheric and stratospheric Psi and EPF Div mean errors. This seems to be not the case as the quantities shown are the global mean vertically averaged mean errors.*

*Please rephrase this sentence so it conveys what you meant originally. "Global average mean errors in the circulation (Fig. 3c,d)... in the stratosphere."*

Thanks for catching this, your interpretation is correct - we've fixed it.

*Page 11, lines 329-331: Perhaps a lat/height cross section figure of the e-folding O3 and CO timescales could be added in the supplement. I wouldn't consider that a must, but it might be helpful to the reader. I don't feel that this is necessary to be shown in the paper; you could just include it in the responses.*

This is a great suggestion, and we've added it as Figure S4 in the (new) supplementary information.

*Page 11, end of line 332: "...Northern Hemisphere" –compared to?*

We've decided to simplify this discussion, so we no longer make this distinction.

*Page 12, line 362: "...are only substantially..." –why only?*

This was poor phrasing on our part, it now reads "are reduced only when using the reference eddy fields." We merely meant that there is only a single case in which the error is appreciably reduced.

*Page 13, line 378: Please note that you still talk about the UTLS as in the mid to upper stratosphere it seems to me that the fluxes do not have the same structure. Also "...over the pole"–clarify which pole?*

We've tweaked this discussion a bit as we have combined the O3/CO sections into one. We now discuss the UTLS and polar middle/upper stratosphere patterns separately (so there is no ambiguity).

*Page 13, line 382: Clarify compared to what CO is increased? Reference climatology or that it is the mean error that increases?*

In the methods section, we now clarify on lines 184-186, "When displaying spatial contour plots, we will always display the mean bias, or difference in climatology, but will always use the SAME to calculate global average mean errors. We will refer to both as the "mean error"."

*Page 14, line 427: Please provide an average estimated range of the stratospheric AoA for the reader.*

Yes, this is a good suggestion, we've added, "Circulation errors integrated over at least the stratospheric age of air, which ranges from one to five years…"

*Typos*

*Page 3, line 60: "...one of none..." –correct to or*

Thank you, corrected.

*Page 3, line 72: "strength" of the "meridional circulation,"*

Thanks!

*Page 15, line 444: "asking sjust" –just*

Thank you, corrected.

**Responses to Reviewer #2**

*Review of "Specified dynamics scheme impacts on wave-mean flow dynamics, convection, and tracer transport in CESM2 (WACCM6)" by Davis et al. In this manuscript, entitled "Specified dynamics scheme impacts on wave-mean flow dynamics, convection, and tracer transport in CESM2 (WACCM6)", the authors present a rigorous examination of the sensitivities of various aspects of the transport and dynamical circulation in CESM2 to nudging the large-scale flow. A systematic exploration of how errors depend on both meteorology frequency as well as nudging timescale is presented. As this manuscript presents an unparalleled level of information, detailing the behavior of the nudging framework within CESM2, it represents an important study that is certainly deserving of publication. I commend the authors for performing this type of evaluation, which is somewhat lacking in the literature and, as illustrated by the authors, riddled with complexities and nuances. There are, however, a few ways in which the manuscript can be improved to both clarify the applicability of these results to other (namely reanalysis) fields as well as to focus the discussion on the major findings in order to communicate more clearly the first order results. As such, my recommendation is "major revisions", but I emphasize to the authors that this study has strong potential and will constitute a valuable contribution to the field, provided that these revisions are incorporated.*

Thank you for your review, we appreciate it!

*Major Comment 1:*

*While I agree with the authors that using meteorology produced from the same underlying GCM is, indeed, the cleanest way to assess the performance of the nudging scheme, one wonders how the major conclusions from the study change (if they do) upon nudging to reanalysis fields. In particular, one of the central challenges with nudging is striking the "correct" balance between sufficiently constraining to the observed (reanalysis) fields, while also not doing too much harm in the actual act of nudging (i.e. producing dynamical inconsistencies in the flow that may further degrade the transport characteristics of the simulation). Therefore, while it is certainly worthwhile to evaluate the performance of the nudging scheme within the (parameter and flow) space of the parent GCM, ultimately what matters in the end is how that scheme operates in the combined "GCM-reanalysis" space. In other words, one needs information (not provided in the current manuscript) about the underlying meteorological biases in the underlying GCM (CESM2). How confident are the authors that a scheme that reproduces CESM2 fields also reproduces other (reanalysis) fields, given what might be quite large model biases? The authors have shown that the behavior of the nudging scheme does not necessarily exhibit nice convergence properties (for example — the minimization of the convective mass flux errors at a 12 hour nudging timescale, with increased errors at both shorter and longer nudging timescales (lines 220-222) ). If the nudging scheme presents such complicated behavior, when constrained with its own fields, one can imagine the behavior might become still more complex when applied to reanalysis fields, potentially leading to major differences in the author' conclusions.*

*To this end, I think a necessary addition to the manuscript is the addition of a few simulations wherein CESM2 is nudged to MERRA-2 (or any reanalysis product of the authors' choosing) using a subset of the nudging timescale parameter combinations that are explored in the original set of experiments. I understand that these caveats are mentioned in lines 431-438 but I am not convinced that this is sufficient to address this issue. I am not suggesting an exhaustive*

*set of runs but, rather, two or three simulations that demonstrate that the main findings of the study also hold when nudging to reanalysis fields. Since CESM2-SD is already set up to nudge to reanalysis fields, I cannot imagine that this is an unreasonable ask.*

We've taken this suggestion seriously and have performed three simulations in which we nudge WACCM to GEOS meteorology (the new nudging scheme requires different pre-processing - it is what was easily available to us). We decided to choose a 12 hour nudging timescale/8 per day meteorology frequency (the maximum possible) as our "best case", and then performed simulations with a 48 hour nudging timescale/8 per day frequency and 12 hour nudging timescale/4 per day meteorology frequency to probe the parameter space. The new results are displayed in Fig. 19 and discussion on lines 445-463.

We find that the temperature and EP flux divergence temporal and spatial errors scale roughly as they do in the simulations in which WACCM is nudged to itself - coarser meteorology frequency and a longer nudging timescale both result in higher error. Similarly, their mean errors increase with longer nudging timescale, which is to be expected in this case because GEOS and WACCM likely have different climatologies.

We do find that the convective mass flux and TEM streamfunction are insensitive to these parameter variations. These convective mass flux errors generally reflect the existing results, but the TEM streamfunction errors do not asymptote as they do when WACCM is nudged to itself. Thus, the added wrinkle with nudging to another model is increased error in the zonal mean meridional circulation, which we note reflects some previous work on this subject.

Overall, we agree that while this adds some extra complexity to our result, we think it still emphasizes that a nudging timescale around 12 hours, with the maximum met frequency possible, is the best configuration using the current iteration of the nudging scheme.

*Major Comment 2:*
*I commend the authors for performing a quite exhaustive examination of the errors in various circulation and transport diagnostics within the context of the simulations considered in this study. At the same time, however, the complexity of the results renders the manuscript very descriptive, and it can be difficult for the reader to extract the key results from some of the more secondary points. In particular, the separate discussions of the ozone and carbon monoxide errors (Sections 7 and 8) are quite long and nuanced. Is there not a way to combine into a single section that is prefaced by a paragraph highlighting the common features among these two constituents (with respect to their response to nudging), followed by a discussion of each regarding the specific characteristics of each field? At present, the number of details presented in the manuscript renders it a bit hard to follow and the authors should better emphasize throughout the main "take-away" messages (which are currently reserved only for the conclusions).*

This is a fair criticism. Having two separate sections made it difficult to compare/contrast the results, so we were leaving it up to the reader to stitch it all together. We've combined these sections into a single section where we can discuss the similarities (and occasional differences). We think this provides the reader with a better appreciation of a key takeaway we did not emphasize enough in the earlier draft - that the two tracers, despite having contrasting production/loss regions, are impacted similarly by transport errors.

*Minor Comments:*
*1. Line 155: Shouldn't the stratospheric vs. tropospheric averages be somewhat latitude dependent, at least enough to distinguish between first-order differences in tropopause height?*

*In particular, my concern is that in the extratropics 200 hPa is already well within the lower stratosphere and yet still quite tropospheric in the tropics. Why not use something a bit more physically based? (i.e. 300 hPa for latitudes > 40S/N and 100 hPa for equatorward latitudes)?*

We initially did not think this would impact the results very much, but we agree there are substantial differences in the error above and below the tropopause, especially for the tracers. We've updated all calculations to use the tropopause pressure, rather than 200 hPa - this did not materially change any results except for the stratospheric ozone mean error, which now scales more cleanly with nudging timescale. Lines 171-173 have been updated to reflect this change.

*2. Figure 4: This is a very important figure, and it is nice to see this result documented so well. It could be worth noting in the text when describing this result that similar behavior was observed in Figure 3b,d in the study indicated below, albeit for only two nudging timescales (and using CAM, nudged to MERRA). Consistent with the results presented in this study, that figure shows that the convective mass fluxes in the 5hr nudged simulation differed substantially more than the fluxes from the 50 hour nudged simulation, relative to MERRA. Perhaps it is worth highlighting this consistency between the two studies.*

*Orbe C., D. W. Waugh, H. Yang, J. F. Lamarque, D. Kinnison, and S. Tilmes (2016), Tropospheric Transport Differences Between Models Using the Same Large-Scale Meteorological Fields, Geophysical Research Letters, 44(2), 1068-1078.*

This was an oversight on our part - we've now added this to the discussion, " (Orbe et al., 2017) similarly show that a CESM1(WACCM4) simulation nudged to MERRA at a 5 hour nudging timescale has a weakened convective mass flux relative to a simulation nudged to MERRA at a 50 hour nudging timescale."

*3. Line 165: "Negative values of the SAME indicate that the field is weaker in magnitude than it is in the reference simulation - up to and including opposite in sign". I do not necessarily agree with this description, and I think the issue is just one of unclear wording. For example, consider that $x\_ref=-2$ and $x\_sd=3$. Then, according to (5), SAME<0. And, yet, the field is \*greater\* in (absolute) magnitude than it is in the reference simulation. Perhaps the "up to an including opposite in sign" just needs to be clarified that it refers explicitly to $x\_sd$.*

Agreed that this phrasing was quite confusing. We've added some detail so that the reader can see the full phase space of interpretations: "Positive values of the SAME greater in magnitude than the reference climatology indicate the field is greater in magnitude and of the same sign. Negative values of the SAME indicate cases where the field is weaker in magnitude than but of the same sign as the reference climatology, to cases where the field is greater in magnitude than and of the opposite sign to the reference climatology."

*4. Line 401: Is conclusion 1(b) true? Don't the errors in convective mass fluxes \*increase\* at nudging timescales shorter and longer than 12 hr?*

We did not intend for readers to interpret the convective mass flux as part of the "circulation" in conclusion 1, but we invited it because was not listed in either! We've now clarified that conclusion 1 applies to the "resolved circulation", and have added "convective mass flux" to conclusion 2.

**Responses to Reviewer 3**

*This paper assesses the impact of various specified dynamics (SD) model configuration parameters on the fidelity of the SD circulation and tracer transport by comparing the SD output to a free-running version of the same model. This paper presents a clever and fairly comprehensive analysis of the sensitivity of the SD fidelity in WACCM6 to nudging timescale and meteorology frequency, and is an important contribution towards better understanding SD model results in general, which have been the source of conflicting results in recent years. As the paper is well written and the analysis sound, I only have a few minor comments for consideration by the authors (listed below).*

Thanks for your review, we appreciate it!

*Line 60: "one of more of" → "one or more of"*

Thank you, we've corrected this.

*Line 75-77: I'd also add that temperature nudging is presumably important for water vapor and cloud microphysics, which is not addressed in this paper.*

That is a great point, and we've added it here and in the conclusion on lines 467-468.

*Lines 96 – 103: In this paper the authors are using an alternative nudging technique that is not the standard CESM configuration as I understand. It would be helpful to know if the results in this paper are in any way sensitive to the nudging methodology. Presumably the authors chose this method because it could apply to different dynamical cores, but it would be helpful for the reader to include some discussion to either state or speculate that how the results presented here may or may not be different than the standard nudging technique. Also related, why did the authors end their sensitivity study at a nudging timescale of 48 hours. I believe the standard CESM SD nudging timescales are longer, like 3 or 5 days. It would seem important to bracket/ include whatever the standard configuration is.*

We've added a clarification on lines 113-116 to address the "SD" compset scheme, " In this formulation, the nudging proactively pulls the model state toward the next instantaneous reference meteorology value. The "SD" scheme calculates xref as a linear interpolation between the two nearest reference meteorology values, which somewhat paradoxically attempts to correct the future state based on present disagreement." While we did not perform any simulations using this older scheme, its mathematical formulation seems suboptimal because it attempts to correct the meteorology based on the current state. Anecdotally, we understand from researchers interested in tropospheric climate and weather that this older scheme may have been trouble for clouds (it may be causing a see-saw around the desired meteorology). However, we do not know of any published analyses that we could cite.

In terms of timescale, we are not aware of any analyses using longer than a 50 hour timescale, which corresponded to a 1% nudging/timestep using the old scheme. All timescales we examined are harmonics of 24 hours; we chose 48 hours as the bracket as a multiple of 24 hours. We think the discussion on lines 142-144 provides some physical interpretation of all of these nudging timescales and the sorts of variability they may constrain.

*Paragraph starting line 96: Are the nudging variables being used instantaneous or averaged output? Please state in the text. Also, I recognize that this may be beyond the scope of this*

*paper, but some discussion/speculation on the impact of using instantaneous vs. averaged fields would be helpful and make this more relevant to real world decisions that modelers make when they are choosing the type of reanalysis output they want to use for input in SD runs.*

That is a good question. We've clarified in the text that we only use instantaneous output by subsampling the every-timestep output, " Here, a free running simulation is used to generate reference meteorology every 30 minute dynamics timestep. For cases other than Nobs = 48 , we subsample this instantaneous output at equally-spaced intervals. While there is some evidence that the use of averaged fields may produce more realistic stratospheric transport (Orbe et al., 2017), it isn't clear whether this would apply to the very high frequency meteorology examined here." We do think it may be the case that this forward-looking nudging scheme may not experience a performance degradation using instantaneous meteorology, but we aren't comfortable making such a statement in the paper without actual results. Certainly, though, the difference between averaged and instantaneous meteorology wanes as the meteorology frequency increases. Taking our suggestion to use the maximum meteorology frequency possible to its limit (which is now 1-hourly ERA5) probably diminishes the role of instantaneous vs. averaged fields.

*Lines ~209 (reference to Figs 1-3): I like that that the authors show both the nudged variable (temperature) as well as derived variables like CMF, streamfunction, etc. However, I am confused as to why zonal and meridional wind errors are not also shown here, as these are the fundamental variables that are being nudged to. For completeness, I suggest adding these to the figures and making more clear that the most basic "test" of an SD model is how well it can reproduce the things it's actually nudging to (i.e., uvT in this case).*

We have added these to supplementary Figures 1-3. Their errors behave almost identically to the corresponding errors in temperature, asymptoting toward zero at sufficiently high meteorology frequency. Our focus on the wave/mean flow decomposition helps to develop a consistent physical understanding of how errors in the circulation project onto errors in transport.

*Lines 210-211: The word nudging is repeated twice*

Thanks, we've corrected this.

*Line 218: If I'm reading Fig. 3 correctly I believe this sentence has it backwards – T, EPF, and TEM streamfunction become biased high by \*decreasing\* nudging timescale, not \*increasing\**

Thanks for catching this - you are correct it should be "decreasing" timescale.

*Line 279: I'm confused why the displaying the negative of error regressions makes the discussion easier.*

We now note on lines 306-307, " We will display the negative of all error regressions - that is, for decreasing nudging timescale and decreasing meteorology frequency." As we hope to quantify the error itself in tracers and tracer transport, we want to focus on the projection that produces error.

*Comment on all figures – I found the choice of differing shades of blue to be somewhat difficult to distinguish on the printed page. I appreciate the need to have a color scale whose hue varies somewhat linearly and intuitively over some range, but am wondering if a two color sequential scale might be a bit more easy to track by eye?*

We are hesitant to use a divergent color scheme because it implies opposite-signed behavior. As a compromise, we've filled in and enlarged the symbols on the line plots so that it is easier to distinguish each curve if the color scheme is insufficient, either rendered or in print.

*Figure 3a – I'm confused as to why the 48h timescale T error looks closest to zero in this figure, whereas in Figures 1a and 2a it looks to have the largest error of any of the nudging timescales. Is there some subtle reason in general why the results in Fig 3 aren't more-or-less a combination of the results in Figs 1 and 2? Might be worth explaining in the text if so. Actually, now that I am looking at it, I realized that I've assumed that Fig. 3. Is the global vertical time average, but I don't see anywhere that this is explicitly stated, so that would be helpful to state clearly in the text and/or caption.*

We've added a clarification to lines 189-190, " In all cases we examine the time-mean of all errors and fields over the entire one year simulation period."

For the mean error, we've added a note to lines 242-243, "At a 48 hour nudging timescale, the nudging is too gentle to have a material impact on the modeled climate". Essentially, if the nudging is very weak, the nudging will not change the average climate, as the climate-timescale processes are sufficiently fast to overcome the nudging. We do agree it is an open question as to why nudging with a sufficiently short timescale leads to a change in the modeled climate; we think it is partly due to the nudging damping convection.